# Global variation in the fraction of leaf nitrogen allocated to photosynthesis

Xiangzhong Luo [1,2,3 ✉], Trevor F. Keenan [2,3 ✉], Jing M. Chen[4], Holly Croft[5], I. Colin Prentice [6,7,8], Nicholas G. Smith [9], Anthony P. Walker [10], Han Wang[7], Rong Wang[4], Chonggang Xu[11] & Yao Zhang [2,3]

Plants invest a considerable amount of leaf nitrogen in the photosynthetic enzyme ribulose-1,5-bisphosphate carboxylase-oxygenase (RuBisCO), forming a strong coupling of nitrogen and photosynthetic capacity. Variability in the nitrogen-photosynthesis relationship indicates different nitrogen use strategies of plants (i.e., the fraction nitrogen allocated to RuBisCO; fLNR), however, the reason for this remains unclear as widely different nitrogen use strategies are adopted in photosynthesis models. Here, we use a comprehensive database of in situ observations, a remote sensing product of leaf chlorophyll and ancillary climate and soil data, to examine the global distribution in fLNR using a random forest model. We find global fLNR is 18.2 ± 6.2%, with its variation largely driven by negative dependence on leaf mass per area and positive dependence on leaf phosphorus. Some climate and soil factors (i.e., light, atmospheric dryness, soil pH, and sand) have considerable positive influences on fLNR regionally. This study provides insight into the nitrogen-photosynthesis relationship of plants globally and an improved understanding of the global distribution of photosynthetic potential.

[1] Department of Geography, National University of Singapore, Singapore, Singapore. [2] Climate and Ecosystem Sciences Division, Lawrence Berkeley National Laboratory, Berkeley, CA, USA. [3] Department of Environmental Science, Policy, and Management, UC Berkeley, CA, USA. [4] Department of Geography and Planning, University of Toronto, Toronto, ON, Canada. [5] Department of Animal and Plant Sciences, University of Sheffield, Sheffield, UK. [6] Department of Biological Sciences, Macquarie University, North Ryde, NSW, Australia. [7] Department of Earth System Science, Tsinghua University, Beijing, China. [8] Department of Life Sciences, Imperial College London, Silwood Park Campus, Ascot, UK. [9] Department of Biological Sciences, Texas Tech University, Lubbock, TX, USA. [10] Environmental Sciences Division and Climate Change Science Institute, Oak Ridge National Laboratory, Oak Ridge, TN, USA. [11] Earth and Environmental Sciences Division, Los Alamos National Laboratory, Los Alamos, NM, USA. ✉email: xzluo.remi@nus.edu.sg; trevorkeenan@berkeley.edu

Plant photosynthesis is the primary means of biochemical energy production and carbon assimilation that supports the growth and survival of most life on Earth[1,2]. The rate at which plants photosynthesize is partly constrained by plant photosynthetic capacity, which largely reflects the active amount and kinetic activity of the ribulose-1,5-bisphosphate carboxylase-oxygenase (RuBisCO) enzyme in leaves[1]. In ecosystem models that simulate plant photosynthesis, the active RuBisCO content is indicated by the temperature-standardized rate of maximum RuBisCO carboxylation ($V_{c_{max}}^{25}$). However, the estimation of $V_{c_{max}}^{25}$ across global scales has been challenging as current approaches provide substantially different estimates[3,4].

RuBisCO is the most abundant protein in the world[5]. Nitrogen invested in RuBisCO makes up a large fraction of nitrogen in leaves, resulting in strong coupling between $V_{c_{max}}^{25}$ and leaf nitrogen content (LNC; area basis)[6]. The fraction of leaf nitrogen allocated to RuBisCO (fLNR) is critical to the variability of the LNC ~ $V_{c_{max}}^{25}$ relationship[7,8], as well as being an indicator of leaf to whole-plant nitrogen use strategies. Several studies have suggested that fLNR is a highly plastic trait, ranging from 9% to 28% across species[9–11]. However, detailed information on the drivers of fLNR variability is lacking, due to difficulties associated with its direct measurement[9,12]. In ecosystem models, fLNR is also often not explicitly considered[4] but see[13].

Many models use empirical linear relationships between $V_{c_{max}}^{25}$ and LNC to estimate $V_{c_{max}}^{25}$, based on their concurrent observations[6,14]. These empirical "nutrient-based" $V_{c_{max}}^{25}$ models use plant functional type (PFT)-specific linear equations to calculate $V_{c_{max}}^{25}$ from LNC, which essentially assume a fixed fLNR per PFT, or in some cases modify fLNR (i.e., proportional to the coefficient of the linear equations) by leaf traits such as LNC or leaf phosphorus content (LPC)[15,16]. Even for models that apply fLNR in the derivation of $V_{c_{max}}^{25}$, fLNR is set as a constant for each PFT[13], similar to the empirical $V_{c_{max}}^{25}$ models[4]. The variation of fLNR within PFTs is largely ignored in such nutrient-based empirical $V_{c_{max}}^{25}$ models. Another class of approaches upscales in situ observations to the globe using linear regressions of $V_{c_{max}}^{25}$ to climate variables[17], under an assumption that local climate constrains leaf photosynthetic traits. This "climate-based" empirical approach provides a higher level of spatial variability for $V_{c_{max}}^{25}$ than nutrient-based approaches[3], and thus a more variable fLNR.

The emergence of optimality hypotheses has motivated the development of new $V_{c_{max}}^{25}$ models that can explicitly or implicitly estimate spatially and temporally dynamic fLNR. For example, an ecological optimality (EO) model based on the coordination[18] and least cost hypotheses[19] predicts $V_{c_{max}}^{25}$ from climate variables, as plants optimize light use efficiency and minimize costs associated with photosynthetic enzymes and water movement in accordance with local weather and climate[20]. This model suggests that plants acclimate to the local climate to optimize carboxylation, and fLNR is implicitly adjusted by the shifting RuBisCO demands for nitrogen[21]. Meanwhile, another optimality model —leaf utilization of nitrogen for assimilation (LUNA)—explicitly suggests that fLNR actively changes with climate under a given LNC, to optimize leaf nitrogen use so as to maximize net photosynthesis[22,23], and it has been incorporated in the latest version of Community Land Model (i.e., v5)[24].

We refer to these two optimality $V_{c_{max}}^{25}$ models characterized by dynamic fLNR as "optimal $V_{c_{max}}^{25}$ models", as opposed to the aforementioned empirical $V_{c_{max}}^{25}$ models. Though empirical models are often statistically derived from the similar ground observations that optimal $V_{c_{max}}^{25}$ models are validated with, or even trained upon, the resulting global distribution of $V_{c_{max}}^{25}$ from these different methods has been reported to be very different[3]. We hypothesize that the stated differences between models reflect their different assumptions regarding fLNR. To test this hypothesis, we need to understand the variability of global fLNR and identify the dominant controls of fLNR at the global scale.

Recent advances in leaf trait compilation and sharing[25–27], machine learning techniques for upscaling[28–31], and remote sensing of leaf traits[32–35] provide a unique opportunity both to create data-driven maps of global $V_{c_{max}}^{25}$ and fLNR, to assess the efficacy of existing estimates. In this study, we first take advantage of newly released data of satellite-derived leaf chlorophyll content[35]—the key pigment in photosynthetic light-harvesting, and a comprehensive dataset of in situ $V_{c_{max}}^{25}$ (8610 obs.; see "Methods") that we compiled from various databases[14,20,25] to infer the global distribution of $V_{c_{max}}^{25}$ using a random forest (RF) approach. We then use the gridded $V_{c_{max}}^{25}$ map and a previously published gridded LNC dataset[28] to generate a data-driven map of global fLNR (see "Methods"). The resulting global distribution of fLNR allows us to examine the dominant controls of global fLNR, and evaluate the nitrogen allocation assumptions of seven competing $V_{c_{max}}^{25}$ models, including four nutrient-based empirical models—EM1 and EM2[6], EM3 and EM4[15], one climate-based empirical model—EM5[17], and two optimality-based models—EO[20] and LUNA[23] (see "Methods").

## Results

We examined the relative importance of 20 environmental (i.e., biotic and abiotic) factors in estimating in situ observations of $V_{c_{max}}^{25}$, and identified remote sensing leaf chlorophyll content (Chl), PFTs, precipitation, and soil pH as the most critical factors (see "Methods"). We used these four predictors in RF to estimate global $V_{c_{max}}^{25}$. The resulting RF $V_{c_{max}}^{25}$ model explained 56.4% of the spatial variance in observed $V_{c_{max}}^{25}$ (Fig. 1a). Global $V_{c_{max}}^{25}$ estimated by RF was $65.7 \pm 13.8\ \mu mol\ m^{-2}\ s^{-1}$ (mean ± sd), ranging from to 27.4 to $158.4\ \mu mol\ m^{-2}\ s^{-1}$ (Fig. 1b). Using RF $V_{c_{max}}^{25}$ as the reference, we found that global $V_{c_{max}}^{25}$ estimated by the seven competing models (i.e., EM1 to EM5, EO and LUNA; see "Methods") showed very different spatial patterns, with the correlation coefficient ($r$) to RF $V_{c_{max}}^{25}$ ranging from −0.14 to 0.30 (Fig. 1c; Supplementary Fig. 1).

The fLNR inferred from RF $V_{c_{max}}^{25}$ and LNC[28] (see "Methods") showed that global vegetation invested 18.2 ± 6.2% of leaf nitrogen in RuBisCO, on par with the value (mean: 17.6%; 90% quantiles: 9.9–26.7%) reported by a previous meta-analysis of leaf chemistry ($n = 138$)[10]. Our RF result suggested that fLNR was very plastic, varying from 4.8% to 59.9% spatially (Fig. 1d). The fLNR of vegetation in boreal (>60°N and S) and tropical zones (30°S–30°N) was generally lower than that in temperate zones (30–60°N and S).

We further evaluated seven $V_{c_{max}}^{25}$ models and found they all demonstrated different implied fLNR patterns than RF fLNR (Supplementary Fig. 2), though the average fLNR reported by the optimal $V_{c_{max}}^{25}$ models (i.e., EO and LUNA) was closer to the average RF fLNR than the empirical models. The fLNR estimated by $V_{c_{max}}^{25}$ models was 13.3 ± 5.3% (EM1), 14.0 ± 4.3% (EM2), 11.3 ± 1.1% (EM3), 6.2 ± 2.2% (EM4), 20.5 ± 11.1% (EM5), 18.9 ± 5.8% (EO) and 20.0 ± 10.9% (LUNA), respectively.

We then examined the changes in fLNR from RF and the other models along with ecological, climate, and soil gradients. According to RF, forests showed significantly ($t$-test; $p < 0.05$) smaller fLNR (15.3 ± 4.7%) than non-forests (20.9 ± 6.2%), consistent with a leaf chemistry analysis[9]. Tropical evergreen forests had less fLNR variability (sd = 2.3%) than other PFTs (from 4.6% for evergreen needleleaf forests to 6.3% for croplands; Fig. 2a).

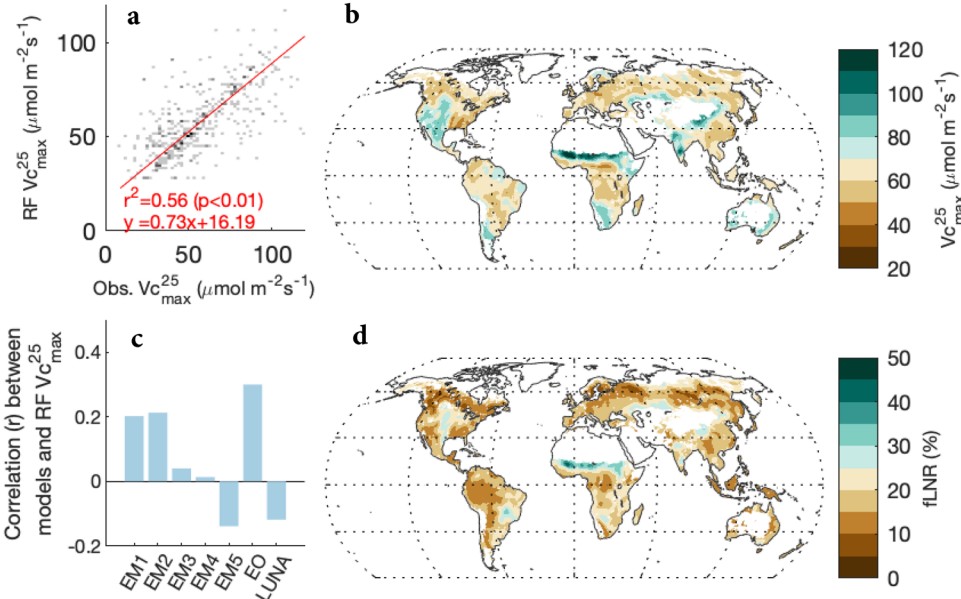

**Fig. 1 Global $V_{c_{max}}^{25}$ and the fraction of leaf nitrogen invested in RuBisCO (fLNR). a** The relationship (type-II regression) between observed in situ $V_{c_{max}}^{25}$ and $V_{c_{max}}^{25}$ estimated by a random forest model (RF) on a 0.5-degree grid; (**b**) the global distribution of $V_{c_{max}}^{25}$ ($\mu$mol m$^{-2}$ s$^{-1}$) estimated by RF using global gridded environmental covariates; (**c**) the spatial correlation between RF $V_{c_{max}}^{25}$ and the estimates of seven competing $V_{c_{max}}^{25}$ models, including five empirical $V_{c_{max}}^{25}$ models (EM1 to EM5) and two optimal $V_{c_{max}}^{25}$ models (EO and LUNA); (**d**) the global distribution of RF fLNR (%). The maps in (**b**) and (**d**) were created by the authors using a Matlab package M_Map (see "Code availability statement").

There was a large variation in RF fLNR within each PFT (Fig. 2a), which indicates that the plasticity of fLNR was dependent on factors other than PFT. This invalidates some empirical $V_{c_{max}}^{25}$ models (i.e., EM1, EM2), which assume fLNR is almost a constant per PFT. EM3 and EM4 modify fLNR by LNC and LPC only, however, such modification was insufficient to account for the observed variation in fLNR (Fig. 2a). EM5, EO, and LUNA demonstrated within-PFT variation in fLNR comparable to that of RF fLNR (Fig. 2a), suggesting that environmental factors adjust within-PFT variability of fLNR.

Since fLNR is critical to the photosynthetic capacity of leaves and there is evidence that the photosynthetic capacity of leaves is dependent on leaf traits[36], climate[14], and soil characteristics[37], we assume that the changes in fLNR can be attributed to these three types of factors. We used a generalized additive model to study the partial response of fLNR to the principal components (PCs) of each group of variables (i.e., leaf traits, climate, and soil; Supplementary Table 1). We found the RF fLNR was most sensitive to the first principal component (PC1) of leaf traits, followed by smaller influence from climate PC1 and soil PC1 (Fig. 2b–d). There was a 25% change in fLNR over the observed leaf traits range. The leaf traits-fLNR relationship suggests a strong constraint of fLNR by the coordination of leaf functional traits (i.e., the leaf economic spectrum)[36]. While the optimal $V_{c_{max}}^{25}$ models (i.e., EO and LUNA) and EM5 captured the response of fLNR to leaf traits PC1 well, they either slightly underestimated or overestimated the sensitivity to leaf traits PC1 (Fig. 2b). Climate PC1 caused RF fLNR to change as much as 10%. However, we found none of the $V_{c_{max}}^{25}$ models captured the observed climate-fLNR relationship, as LUNA underestimated the sensitivity of fLNR to climate PC1, and EM5 and EO suggested the opposite direction of response (Fig. 2c). The influence of soil PC1 on RF fLNR was less than 3%. All optimal $V_{c_{max}}^{25}$ models detected a similar magnitude of the soil PC1 effect on fLNR but showed slightly different response curves (Fig. 2d).

Globally, we found that fLNR was predominately determined by its relationship with other leaf traits, as on 63.5% of the

vegetated land surface leaf traits explained more than 50% of the changes in fLNR, while climate and soil characteristics imposed substantial impacts at the regional scale (Fig. 3a). Climate explained 30% or more of the changes in leaf fLNR over 50.9% of the vegetated land surface, including the southeastern U.S., Central America, Eastern Europe, tropical and southern Africa, a large part of China, India, and Australia. Meanwhile, soil explained 15% or more of the changes in leaf fLNR on 11.9% of the vegetated land surface, including western and eastern North America, northern Eurasia, and temperate Africa (Fig. 3a). The average changes in fLNR driven by leaf traits, climate and soil were 4.3 ± 2.7, 2.1 ± 1.7%, and 0.8 ± 0.6%, respectively.

We further quantified the influence of each variable on fLNR using a multivariate linear model (see "Methods"). Our result suggested that leaves with larger leaf mass per area (LMA) have smaller fLNR (Fig. 3b), indicating that non-photosynthetic proteins and amino acids used an increasingly larger portion of leaf nitrogen. A 1 g/m$^2$ increase in LMA caused a 0.19 ± 0.001% (95% CI) decrease in fLNR. Meanwhile, LPC was positively related to fLNR, with a 0.1 g/m$^2$ increase in LPC leading to a 4.22 ± 0.047% increase in fLNR. The response of fLNR to LPC for tropical evergreen forests and mixed forests was stronger than other PFTs (Fig. 3c, Supplementary Table 2). Changes in fLNR were positively correlated to vapor pressure deficit (VPD). An increase of 0.1 kPa in VPD drove fLNR to increase by 0.48 ± 0.002%. Air temperature, precipitation, and soil water content overall showed a negligible impact on fLNR. The fLNR of non-forest ecosystems (i.e., croplands, shrublands, and wetlands) increased with photosynthetically active radiation (PAR), suggesting plants enhance nitrogen use under high light conditions. Overall, a 100 $\mu$mol/m$^2$/ s increase in PAR led to a 0.26 ± 0.001% increase in fLNR. Finally, soil pH and soil sand percentage were identified as the two most important soil factors influencing fLNR. fLNR increased by 0.25 ± 0.03% and 0.03 ± 0.0002% in response to a unit increase in pH (unitless) and soil sand percentage (%), respectively (Fig. 3c). In addition, some soil characteristics demonstrated PFT-specific influence on fLNR, notably a positive effect of soil bulk density on

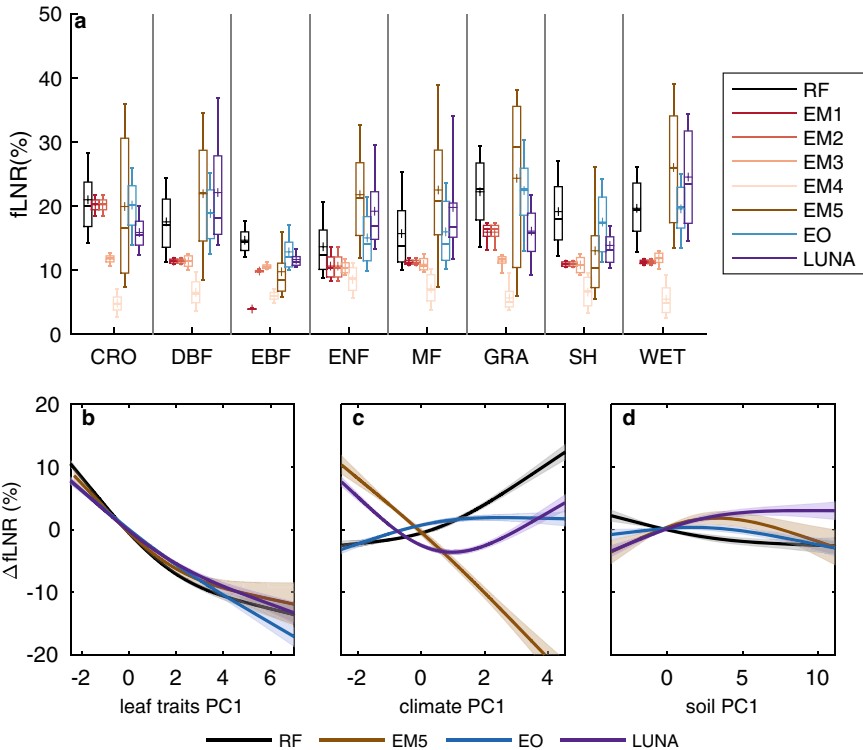

**Fig. 2 Comparison of the fraction of leaf nitrogen in RuBisCO (fLNR) estimated by $V_{c_{max}}^{25}$ models to fLNR estimated by random forest (RF).** Five empirical $V_{c_{max}}^{25}$ models (EM1 to EM5) and two optimal $V_{c_{max}}^{25}$ models (EO and LUNA) were examined. **a** fLNR per plant functional type (PFT) and the responses of fLNR to the first principal components (PC1s) of (**b**) leaf traits, (**c**) climate variables, and (**d**) soil variables. In (**a**), the acronyms and the numbers of half-degree cells for PFTs are: cropland (CRO; n = 10,525), deciduous broadleaf forest (DBF; n = 7525), evergreen broadleaf forest (EBF; n = 4626), evergreen needleleaf forest (ENF; n = 6259), mixed forest (MF; n = 713), grassland (GRA; n = 5771), shrubland (SH; n = 3606) and wetland (WET; n = 940). For each box plot, the cross indicates the mean, the center line indicates the median, the box indicates the upper and lower quartiles and the whiskers indicate the 10th and 90th percentiles of the data. In (**b**)–(**d**), Y axis indicates the partial changes in fLNR (ΔfLNR; unit: %). PC1s account for 83.4%, 57.3%, and 60.0% of the variance in leaf traits, climate variables, and soil variables, respectively (Supplementary Table 1). The partial response of fLNR to PC1s is acquired using a generalized additive model where the solid line indicates the mean partial response, and the shadings indicate one standard error augmented by 10.

fLNR of croplands and grasslands, and a negative effect of soil silt percentage on fLNR of evergreen needleleaf forests.

## Discussion

In this study, we produced a global fLNR and $V_{c_{max}}^{25}$ map using an RF model trained primarily by remote sensing and in situ observations and examined seven $V_{c_{max}}^{25}$ models based on 5 competing hypotheses with regard to their assumptions on fLNR. Our results suggested that the global average fLNR was 18.2 ± 6.2%, and the global distribution of fLNR was dominated by the interaction between fLNR and leaf traits (i.e., LMA and LPC), followed by regional influences from climate (i.e., VPD and PAR) and soil characteristics (i.e., soil pH and sand percentage). We used RF fLNR distribution and its relationships with environmental covariates to evaluate five empirical and two optimal $V_{c_{max}}^{25}$ models, and found that the models showed different degrees of inefficacy in reproducing RF fLNR. Here, we discuss the mechanisms underlying the detected fLNR responses to leaf traits, climate, and soil characteristics and propose future directions to improve the simulation of fLNR and $V_{c_{max}}^{25}$ in models.

### Negative correlation between fLNR and LMA. Our finding that fLNR is negatively related to LMA agrees with a previous meta-analysis that found fLNR decreases by 0.54 ± 0.08% with a 1 g/m² increase in LMA based on a univariate regression[10], though another study reported that the negative relationship between fLNR and LMA was non-significant using a smaller dataset[11].

Using the global dataset, we found a relatively small sensitivity of fLNR to LMA (−0.19 ± 0.001% per 1 g/m²) when accounting for climate and soil (Fig. 3b).

Higher LMA is the result of plants allocating more biomass and nitrogen to building cell walls, which may cause a reduction in $CO_2$ diffusion into the mesophyll as well as relative nitrogen allocated to RuBisCO[38]. Leaves with greater LMA are tougher and usually have a longer leaf lifespan[11,36]. Therefore, the negative correlation between fLNR and LMA highlights the trade-off between photosynthesis and persistence along the leaf economic spectrum: on one end, leaves invest more nitrogen in RuBisCO to increase the photosynthetic capacity and enhance carbon uptake; on the other end leaves invest more nitrogen in structural biomass to improve leaf longevity and lengthen the carbon uptake period. The latter is especially true for evergreen species that have greater LMA and smaller fLNR than deciduous and herbaceous species[10]. The coordination of fLNR and LMA is also consistent with a recent analysis highlighting the role of LMA in determining the variation and predictability of LNC in ecosystem models[39].

In addition, we found that LPC increases fLNR in tropical evergreen forests and mixed forests, which tend to be more phosphorus limited[40]. Our result is consistent with previous studies reporting coupled leaf photosynthetic capacity (i.e., $V_{c_{max}}^{25}$ or maximum photosynthetic capacity ($A_{max}$)) and LPC for tropical species[41,42]. This result indicates potential widespread adjustments of plants nitrogen use by phosphorus investment for

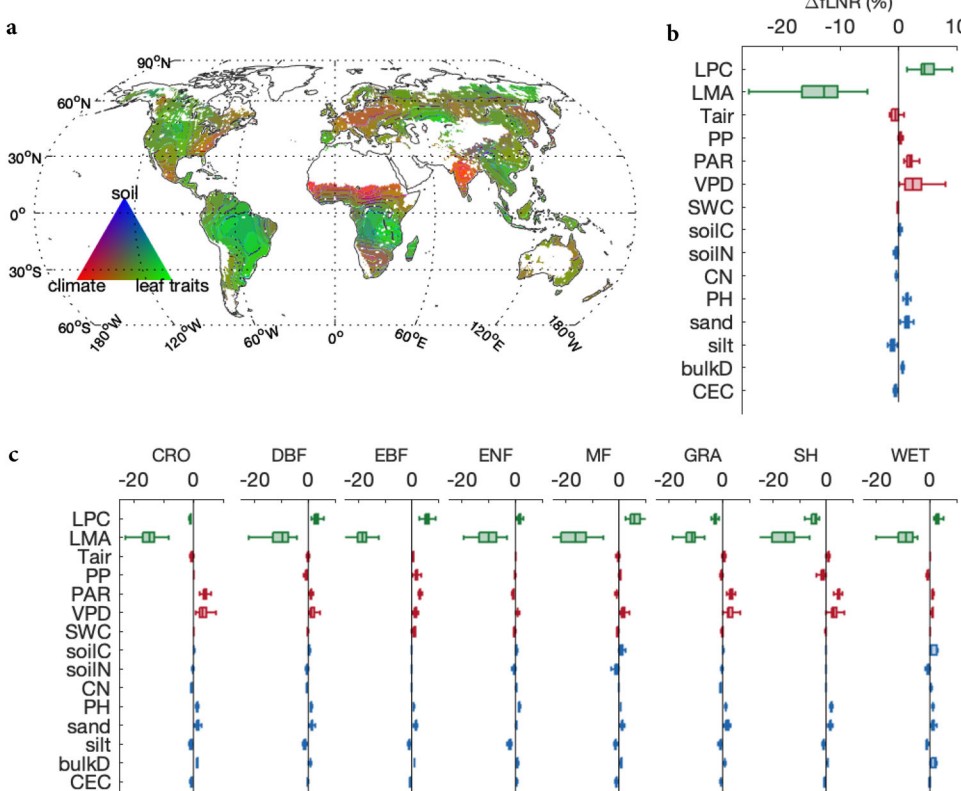

**Fig. 3 The influences of leaf traits, climate, and soil variables on fLNR. a** Dominant factors controlling fLNR changes over the globe; (**b**) the changes in global fLNR attributed to different variables; (**c**) the changes in PFT-specific fLNR attributed to different variables. In (**a**), the percentage contributions from climate and soil are augmented by 50% for demonstration purposes. The variables are leaf phosphorus content (LPC), leaf mass per area (LMA), annual mean air temperature (Tair), precipitation (PP), photosynthetic active radiation (PAR), vapor pressure deficit (VPD), soil water content (SWC), soil organic carbon content (soilC), total nitrogen content (soilN), the ratio of soilC to soilN (CN), soil pH, soil sand and silt percentage (sand and silt), bulk density (bulkD) and cation exchange capacity (CEC). For each box plot in (**b**) and (**c**), the centerline indicates the median, the box indicates the upper and lower quartiles and the whiskers indicate 1.5 times the interquartile range away from the top or bottom of the box. The numbers of samples for box plot in (**b**) and panels in (**c**) are 39147, 10354, 7467, 4602, 6069, 701, 5548, 3534, and 872, respectively. The map in (**a**) was created by the authors using a Matlab package M_Map (see "Code availability statement").

photosynthesis and plant growth[43] in tropical and mixed forests. In addition, we note that the productivity of some grasslands[44,45] and boreal forests[46,47] has also been reported to be limited by phosphorus availability, however, we did not detect a strong positive dependence of fLNR on LPC globally for these ecosystems in our study. The difference potentially suggests that the phosphorus limitation of grasslands and boreal forests is not as prevalent as that for tropical and mixed forests (though some mixed forests are in the boreal region).

**Climate and soil impacts on fLNR.** The response of fLNR to climate is often implicitly included in $V_{c_{max}}^{25}$ models. We found that fLNR was sensitive to annual VPD globally. Several studies have reported that plants in arid environments (i.e., high VPD) tend to have a higher $A_{max}$ and LNC[48,49] as plants enhance photosynthetic capacity to maintain a given assimilation rate with lower stomatal conductance and reduced water loss. Such a response to aridity has been described using the least-cost theory[19,21]. Our results show that other than $A_{max}$ and LNC, fLNR also increases with VPD, consistent with a recent study reporting higher nutrient use efficiency for plants in semi-arid ecosystems of the African Sahel[49]. We note that an earlier study reporting differently that a dry site has a smaller $V_{c_{max}}^{25}$/LNC ratio (i.e., smaller fLNR) than a wet site[19], though it used annual precipitation, not VPD to define aridity.

In addition, the positive relationship between PAR and fLNR for non-forests (Fig. 3c) provides a potential explanation of the light acclimation of photosynthesis, as several studies have found that leaf and ecosystem $A_{max}$ can be enhanced by intermediate to long-term average PAR[50–52]. For non-forest ecosystems, our results suggest that photosynthetic light acclimation emerges as plants increase fLNR in response to increasing annual PAR. However, for forests (except EBF) the results suggest that photosynthetic light acclimation may emerge more due to the increase in LNC as we did not detect a positive response of fLNR to light (Fig. 3c).

Soil characteristics have been reported to influence $A_{max}$ and LNC[37], but we found no studies that have examined the impact of soil characteristics on fLNR. Among the eight soil properties we examined, we found positive responses of fLNR to soil pH and soil sand percentage, followed by small influences of bulk density and silt for certain ecosystems (i.e., croplands, needle leaf forests). pH influences the ability of soil to hold on to nutrients, including $Ca^{2+}$, $K^{2+}$, and $Mg^{2+}$, that are essential to plant growth. A higher pH means more available nutrient cations as acid soils replace nutrient cations with $H^+$. Several studies have reported a positive effect of pH on $A_{max}$[37], non-temperature standardized $V_{c_{max}}$[20], and LNC[39]. Soil sand percentage had a positive impact on fLNR, possibly because sandy soils tend to be less fertile[53] and thus stimulate plants to use their nitrogen more efficiently for photosynthesis and growth. The global influence of soil on fLNR

was generally smaller than leaf traits and climate, but our analysis indicated that on 11.9% of the vegetated surface, soil characteristics contributed more than 15% of the changes in fLNR (Fig. 3a).

Notably, our study found that the soil nitrogen content has a limited impact on the spatial variation of fLNR (Fig. 3). The result implies that processes such as nitrogen deposition/addition are unlikely to affect plants fLNR. The soil nitrogen map we used was upscaled from ground observations of soil profiles in the World Soil Information Service (WoSIS) database. About 47.4–81.4% of the soil profiles in WoSIS are collected from the 1980s to 2020s[54], when there were strong N deposition effects[55]. Therefore, we expect the N deposition effect has been implicitly included in our analysis. We acknowledge that some studies have suggested N deposition influenced leaf nitrogen content and photosynthesis[56,57], however, the influence is limited to certain biomes, deposition load range, and time after the deposition. It is unclear whether these localized and time-dependent effects can influence the global variation of fLNR.

**Uncertainty in the derivation of fLNR and $V_{c_{max}}$.** fLNR was derived based on Eq. (1) (see "Methods") that mechanistically links $V_{c_{max}}^{25}$, LNC, and fLNR, with the assumption that specific activity of RuBisCO ($\alpha^{25}$) and mass ratio of RuBisCO to nitrogen (fNR) are relatively constant values. The average uncertainty of RF fLNR was about $4.20 \pm 2.20\%$ (Supplementary Fig. 3). The uncertainty of fLNR was propagated from several sources including RF $V_{c_{max}}^{25}$, $\alpha^{25}$, fNR, and LNC (Supplementary Fig. 3). Among them, the $\alpha^{25}$ ranges between 47.34 and 60 μmol $CO_2$/g RuBisCO/s, and fNR ranges between 6.11 and 7.16 g RuBisCO/g N[4]. Our uncertainty test showed that the influence of $\alpha^{25}$ and fNR uncertainties on global fLNR were only around $1.13 \pm 0.39\%$ and $0.80 \pm 0.27\%$, respectively (see "Methods"; Supplementary Fig. 3). Physiologically, $\alpha^{25}$ is a value that reflects the change in active sites of RuBisCO and the kinetic constant of the enzyme RuBisCO ($k^{25}$). The number of active sites of RuBisCO is often regarded as a fixed value (set at $6 \times 10^{23}$/mol RuBisCO) for vegetation on the land surface[5], but there are reports showing that $k^{25}$ varies with species[9], leaf ages[58], and temperature[59]. While these dependencies are elusive due to limited observations, previous studies have reported that $k^{25}$ negatively correlates with LNC[60] and LMA[61]. The negative relationship between $k^{25}$ and LMA or LNC is potentially caused by the relatively lower drawdown of $CO_2$ from intercellular spaces to the chloroplast as increased LMA increases mesophyll resistance. In that case, the negative dependence of $k^{25}$ and $\alpha^{25}$ on LNC and LMA might account for part of the negative dependence of fLNR on LMA that we found (Fig. 3b), though the negative influence of LMA on $\alpha^{25}$ was weak and within the range of uncertainty, we quantified (Supplementary Fig. 3).

Compared to $\alpha^{25}$ and fNR, the uncertainties in LNC and RF $V_{c_{max}}^{25}$ incurred larger uncertainties in fLNR. We found that LNC alone caused changes of $3.35 \pm 2.16\%$ in fLNR and RF $V_{c_{max}}^{25}$ caused $3.13 \pm 1.50\%$ (Supplementary Fig. 3). Our study is the first attempt to upscale in situ $V_{c_{max}}^{25}$ to the globe using remote sensing, while similar studies have done that for other leaf traits[33]. The observations used for training RF were densely distributed in Europe and North America, while inner Asia, Southeast Asia, Africa, and high-latitude regions are much less constrained by observations (Supplementary Fig. 6a). In addition, we did not consider temperature acclimation when standardizing in situ $V_{c_{max}}$ to $V_{c_{max}}^{25}$ (Eq. (2)), in order to facilitate the comparison with models that only estimate $V_{c_{max}}^{25}$. However, the uncertainty related to temperature scaling should be limited as acclimated and non-

acclimated temperature scaling factors for $V_{c_{max}}$ are similar under 30 °C[62,63].

The choice of an LNC map is another source of uncertainty in the derivation of fLNR. There are several global LNC maps available other than the EB17[28] map we used, namely AMM18[33] and CB20[31]. Each product has been validated in their respective studies (Supplementary Table 3). To examine the uncertainty incurred by the choice of LNC maps, we calculated fLNR using each of the three LNC maps. The three resulting fLNR maps show similar spatial patterns (Supplementary Fig. 10), with the spatial correlation coefficients ($r$) between them ranging from 0.57 to 0.71 ($p < 0.01$). Examining the influences of environmental variables on fLNR, we found the fLNR based on EB17 and AMM18 demonstrated similar results—fLNR was primarily influenced by LMA, LPC, VPD, PAR, and soil pH. Meanwhile, the fLNR based on CB20 was mostly influenced by soil pH, LMA, VPD, air temperature, and soil sand percentage. Noting that the CB20 LNC map has lower $R^2$ in its cross-validation compared to that of EB17 and AMM18 (Supplementary Table 3), we have more confidence in the fLNR maps based on EB17 and AMM18. In our study, we used EB17 as the principal LNC map since it demonstrated the highest $R^2$ in validation (Supplementary Table 3) and it was more consistent with the other two LNC maps than AMM18 (Supplementary Fig. 9). We acknowledge that the AMM18 LNC map has a smaller RMSE in validation compared to EB17 though it has a slightly lower $R^2$ (Supplementary Table 3). In this study, we do not identify which LNC map is more accurate but show that the choice between EB17 and AMM18 has a limited influence on our conclusion regarding the dominant controls for fLNR (Supplementary Fig. 10g, h).

**Implication for modeling photosynthesis in ecosystem models.** The accurate simulation of fLNR provides a reliable constraint on vegetation photosynthesis, though fLNR is often not an explicit variable in $V_{c_{max}}^{25}$ and ecosystem models[4]. Our results suggest that the conventional PFT-specific method is not effective due to the large variation in fLNR within PFTs. The development of optimal $V_{c_{max}}^{25}$ models is a step forward in estimating spatially varying fLNR, though they currently also show some degree of inaccuracies. We propose two directions moving forward:

(1) Improve optimal $V_{c_{max}}^{25}$ models. We found that optimality models demonstrated some promising strengths, e.g., LUNA and EO detected the coordination between leaf traits and fLNR, however, they demonstrated different responses to climate and soil. The difference between LUNA and EO fLNR is perhaps due to two reasons: (a) the different optimization approach taken and (b) the representativeness of training and validation datasets.

First, the EO model estimates photosynthetic capacity based on the first principles of photosynthesis, that plants minimize the relative carbon and water costs of photosynthesis per unit carbon assimilated while coordinating the electron transport rate-limited and RuBisCO-limited rates of photosynthesis to maximize photosynthesis while minimizing enzymatic and water costs[20]. While the LUNA model dynamically adjusts the fraction of nitrogen invested in different components of photosynthetic and metabolic processes in leaves (i.e., light capture, electron transport, carboxylation, and respiration) to maximize net photosynthesis (gross photosynthesis—photorespiration —dark respiration)[22,23]. LUNA and EO adopt cost functions that are formulated by different biological and environmental constraints to reach also slightly different goals of carbon gain (see "Methods").

Second, our results suggest that subsets of global $V_{c_{max}}^{25}$ demonstrate very different sensitivities to biological and environmental factors (Supplementary Fig. 5). For example, the $V_{c_{max}}^{25}$

dataset used to parameterize the LUNA model identified PFT, LMA, LPC, and cation exchange capacity as the critical factors for $V_{c_{max}}^{25}$, while the NS19 dataset[20] was used to validate the EO model identified PFT, LNC, soil water content, and air temperature (Supplementary Fig. 5). The dataset we compiled combines the NS19, TRY, and LUNA datasets, and shows that leaf chlorophyll content, PFT, precipitation, and soil pH are the most important factors. Therefore, the choice of a subset for training model parameters might result in model biases—e.g., the underperformance of LUNA was likely caused by the relatively small dataset that was used for its parameterization rather than inherent issues of its structure. This thus suggests that using a more representative dataset of $V_{c_{max}}^{25}$, such as the one used here, might improve optimal $V_{c_{max}}^{25}$ model performance.

(2) Estimate fLNR empirically using key predictors. Based on the dominant controls we identified for fLNR, it is feasible to develop an empirical equation to estimate fLNR and then $V_{c_{max}}^{25}$. fLNR is primarily determined by the coordination of leaf traits, which is represented by a negative relationship between fLNR and LMA, and a positive relationship between fLNR and LPC. The effects of VPD, PAR, soil pH, and sand also need to be considered (Fig. 3b). By adopting a multivariate linear model, we obtained fLNR = −0.19LMA + 42.2LPC + 4.76VPD + 0.25 pH + 0.0026PAR + 0.032sand + 2.4 for global vegetation and specific fLNR equations for each PFT (see "Methods"; Supplementary Table 2). In fact, EM3 and EM4 have attempted to use LNC and LPC to adjust fLNR[15], while the low-biased fLNR from EM3 and EM4 might have resulted from the relatively small dataset of phosphorus-limited observations used to derive the model. Our results suggest that empirical $V_{c_{max}}^{25}$ models can be improved by including climate and soil factors.

In addition, the data-driven $V_{c_{max}}^{25}$ map provides a direct constraint on the spatial variations of vegetation photosynthetic capacity. We release the $V_{c_{max}}^{25}$ map and its associated uncertainty (i.e., one standard deviation of estimates from bagged trees in the RF) to facilitate large-scale ecological and modeling studies (Supplementary Fig. 8). Since the remote sensing retrieval of leaf chlorophyll content is for top leaves[35], and the phenological stages of trait samples are not often available, the $V_{c_{max}}^{25}$ estimated in our study can be interpreted as a multi-year average $V_{c_{max}}^{25}$ for top canopy leaves. The seasonality and the within-canopy variations of $V_{c_{max}}^{25}$ are not accounted for in the map.

In conclusion, we have used observations from a range of sources to develop data-driven global maps of $V_{c_{max}}^{25}$ and fLNR, which are critical to understanding vegetation nitrogen use and reducing the uncertainty in estimates of photosynthetic carbon assimilation. We find that the global distribution of fLNR is largely determined by LMA and LPC, in concordance with the leaf economic spectrum, as well as regional climate (i.e., VPD and PAR) and soil properties (i.e., pH and sand fraction). The new understanding and data presented in this study allow for future benchmarking and improvement of $V_{c_{max}}^{25}$ and photosynthesis models, and provide insight into nitrogen use strategies of global vegetation.

## Methods

**Derivation of the fLNR map.** The fraction of leaf nitrogen invested in RuBisCO (fLNR) is related to the photosynthetic capacity of leaves, which is indicated by the maximum carboxylation rate normalized to 25 °C ($V_{c_{max}}^{25}$). Direct measurements of fLNR are sparse, however, as they require the chemical extraction of RuBisCO – a soluble protein[9,12]. We, therefore, used Eq. (1)[8,13] to indirectly derive fLNR using concurrent $V_{c_{max}}^{25}$ (μmol $CO_2$/m²/s) and leaf nitrogen content (LNC):

$$V_{c_{max}}^{25} = \alpha^{25} \times LNC \times fNR \times fLNR \qquad (1)$$

where $\alpha^{25}$ is the specific activity of RuBisCO, that is, the maximum rate of RuBP carboxylation per unit RuBisCO protein (47.34 μmol $CO_2$ /g RuBisCO/s), LNC is area-based leaf nitrogen content (g N/m²) estimated from data in a published study[28], fNR is the mass ratio of RuBisCO molecule to N in RuBisCO molecule (6.25 g RuBisCO/g N)[4]. The uncertainty of fLNR incurred by these parameters was quantified in Supplementary Fig. 3.

In our study, the LNC data we used to derive fLNR was acquired from EB17[28]. EB17 provides data-driven estimates of global mass-based leaf nitrogen content (LNCm; unit: mg/g), specific leaf area (SLA; unit: m²/kg), and their associated uncertainties. We estimated LNC using the equation LNC = LNCm/SLA (unit: g/ m²). The uncertainty of LNC was propagated from the uncertainties of LNCm and SLA generated from 1000 bootstrapping tests. We note there are two alternative global LNC maps available: AMM18[33] and CB20[31]. Comparing the three LNC maps (Supplementary Fig. 9), we found the spatial patterns of EB17 and AMM18 are similar to each other, with EB17 reported a relatively larger spatial gradient of LNC. CB20 LNC shows less evident spatial variation than EB17 and AMM18, and it has the lowest R² in validation among the three products (Supplementary Table 3). Since EB17 has the highest R² in validation among the three, we chose the EB17 LNC map as the principal LNC in our analysis, but have also examined the impacts of using alternative LNC maps on our results (Supplementary Fig. 10).

We first acquired in situ observations of $V_{c_{max}}^{25}$ of C3 species from multiple sources, including the TRY database version 5[25] and the NS19 dataset[20], which contains the training dataset for the LUNA model[14]. We merged these datasets and removed the overlapping records to obtain independent records ($n = 8610$; Supplementary Fig. 6a), where 12.9% of samples were deciduous tree species, 13.7% were evergreen tree species and 73.3% were herbaceous species.

We then used a peaked Arrhenius function to standardize $V_{c_{max}}$ to 25 °C[62]:

$$Vc_{max} = Vc_{max}^{25} \exp[H_a(T_l - T_{ref})/(T_{ref}RT_l)] \frac{1 + \exp\left(\frac{T_{ref}\Delta S - H_d}{T_{ref}R}\right)}{1 + \exp\left(\frac{T_l\Delta S - H_d}{T_lR}\right)} \qquad (2)$$

where $T_l$ is the growing temperature of leaf in Kelvin, $T_{ref}$ is the reference temperature of 298.15 K, $H_a$ is the activation energy for carboxylation (71513 J mol⁻¹), $H_d$ is the deactivation energy (200,000 J mol⁻¹), $\Delta S$ is an entropy term (649.12 J mol⁻¹ K⁻¹) and R is the universal gas constant (8.314 J mol⁻¹ K⁻¹).

We then used an RF model trained on the 8610 samples and several predictors to estimate gridded $V_{c_{max}}^{25}$. The RF approach is selected as (1) the tree-based method is suited to dealing with categorical variables, such as PFTs and climate zones, that are often used in global extrapolation, and (2) the tree-based method is compatible with *the prediction of variable importance by permutation*, which provides a useful way to reduce the number of variables for training and avoiding overfitting[64]. We did not apply species/genus abundance weights to trait values for training and validation purposes, because (1) the available species abundance information at the community level for $V_{c_{max}}^{25}$ was not the same as that for other leaf traits (i.e., leaf nitrogen content) that are potential predictors; (2) we had a much smaller number of samples for training if we aggregated trait values to the community-level (from $n = 8610$ to $n = 429$), which led to a greater risk of overfitting the RF model.

We chose 20 candidate predictors, including LNC (unit: g/m²), LPC (unit: g/ m²), and LMA (unit: g/m²) acquired from a previously published upscaling study[28], leaf chlorophyll content of top leaves (Chl; unit: μg/cm²) derived from remote sensing[35], PFTs from the global land cover map produced by the Climate Change Initiative (CCI) of the European Space Agency, annual mean air temperature (Tair; unit: °C), precipitation (PP; unit: mm/year), photosynthetic active radiation (PAR; unit: μmol/m²/s), vapor pressure deficit (VPD; unit: kPa) from the Climate Research Unit (CRU) TS4.01[65], annual mean soil water content (SWC; unit: m³/m³) and alpha (i.e., evapotranspiration/potential evapotranspiration; unitless) calculated from a bucket model (SPLASH) using CRU TS4.01[66], Koeppen climate classification[67], and soil organic carbon (soilC; unit: g/ m³), total nitrogen (soilN; unit: g/m³), carbon to nitrogen ratio (CN), soil pH, percentage of sand, percentage of silt, bulk density (bulkD; unit: g/cm³) and cation exchange rate (CEC; unit cmol/kg) obtained from the SoilGrids project[30]. The PFTs include croplands (CRO), deciduous broadleaf forests (DBF), evergreen broadleaf forests (EBF), evergreen needleleaf forests (ENF), mixed forests (MF), shrublands (SH), grasslands (GRA), and wetlands (WET).

To construct the RF, we first trained 200 bagged decision trees using all 20 predictors and quantified the importance of each predictor for in situ $V_{c_{max}}^{25}$ estimation (Supplementary Fig. 5). We fed predictors one-by-one to the RF, which consisted of 200 bagged decision trees, based on the importance rankings of predictors until the out-of-bag (OOB; similar to drop-one bootstrapping) test of the RF explanatory power ($r^2$ between the estimates and observations) no longer increased and the OOB error (mean squared error between the estimates and observations) no longer decreased. This process selected 4 important predictors out of the 20 predictors (remote sensing-based Chl, PFT, PP, and soil pH; Supplementary Fig. 4a) as input for the final RF.

We trained the RF using the top 4 predictors and used both conventional and spatial cross-validation to examine the reliability of the RF. We used 80% of samples for training and 20% for validation. The spatial cross-validation is similar to conventional cross-validation, but we removed the validation data points within 1.5° (~150 km) of the training samples to avoid spatial autocorrelation[68]. The

spatial cross-validation showed an $R^2 = 0.52 \pm 0.36$ and an RMSE of 39.9 ± 14.5 µmol m$^{-2}$ s$^{-1}$, while the conventional cross-validation showed an $R^2 = 0.52 \pm 0.01$ and an RMSE of 20.7 ± 0.3 µmol m$^{-2}$ s$^{-1}$ (Supplementary Fig. 11). The spatial cross-validation showed more variable $R^2$ and larger RMSE than the conventional cross-validation as the former has less samples for validation after removing the spatially autocorrelated samples, and those spatially autocorrelated samples generally have small RMSEs between the observations and the estimates. The accuracy of the conventional cross-validation is comparable to previous trait upscaling studies[28,31,33,69] (also see Supplementary Table 3). After validating the RF, we used gridded predictors and the trained RF to extrapolate a global map of $V_{c_{max}}^{25}$. We quantified the uncertainty of RF $V_{c_{max}}^{25}$ as the standard deviation of the estimates from bagged trees in RF.

Finally, we applied Eq. (1) to get global fLNR from RF $V_{c_{max}}^{25}$ and gridded LNC and quantified its associated uncertainty. The uncertainty of fLNR comes from four sources: uncertainties in $V_{c_{max}}^{25}$, LNC, $\alpha^{25}$, and fNR. The uncertainty of $V_{c_{max}}^{25}$ was quantified in training RF, the uncertainty of LNC was provided in the original gridded data[28]. According to the literature, $\alpha^{25}$ ranges from 47.3 to 60.0 µmol CO$_2$ /g RuBisCO/s, and fNR ranges from 6.11 to 7.16 g RuBisCO/g N[4]. We evaluated the impacts of these different sources of uncertainty using 1000 bootstrapping tests, each test using a random $V_{c_{max}}^{25}$, LNC, $\alpha^{25}$, and fNR within their uncertainty ranges (Supplementary Fig. 3).

**Attribution of fLNR changes to ecological and climate factors**. Since fLNR is critical to the photosynthetic capacity of leaves, and there is evidence that the photosynthetic capacity of leaves is dependent on leaf traits[36], climate[14], and soil properties[37], we performed an analysis to attribute the changes in fLNR to these variables. We first divided the variables into three groups: leaf traits, climate, and soil, and conducted a principal component analysis (PCA) on each group, in order to remove the correlation between variables in each group. The explained variance of PCs and the loadings of variables are in Supplementary Table 1. All variables were normalized before the PCA analysis.

We used the first three principal components (PC1, PC2, and PC3) from each group and dummy variables defined by PFTs to fit fLNR in a generalized additive model (GAM) to quantify the partial and non-linear response of fLNR to PCs. Based on the summations of changes in fLNR (ΔfLNR) to leaf traits PCs, climate PCs, and soil PCs, we quantified the fLNR attributable to leaf traits (excluding LNC as it has been used to calculate fLNR), climate and soil, and calculated their relative importance to changes in fLNR over the globe. In this study, we first focused on the response of fLNR to PC1s, as PC1s explained 83.4%, 57.3%, and 60% of the variances of leaf traits, climate, and soil, respectively. Then, we summed the responses of fLNR to the top three PCs (~90% of the variances explained) of each group (i.e., leaf traits, climate, soil), and regarded the sums of the responses of the top three PCs as the total effects of leaf traits, climate, and soil on fLNR. Finally, we used multivariate linear models to fit variables in each group to their corresponding total group effect, and in such a way, we obtained the sensitivity of fLNR to individual variables (Supplementary Table 2).

**Assumptions of fLNR in $V_{c_{max}}^{25}$ models**. Each $V_{c_{max}}^{25}$ model has an underlying assumption of the nitrogen use strategy of plants (i.e., fLNR), which can be inferred using Eq. (1). We derived the fLNR of five empirical $V_{c_{max}}^{25}$ models (EM1 to EM5) and two optimal $V_{c_{max}}^{25}$ models (EO and LUNA) (Supplementary Fig. 2). To derive $V_{c_{max}}^{25}$, we used the same CRU TS4.01 climate[65] and the gridded leaf traits dataset EB17[28] for all models.

*EM1 and EM2*[6]. The two models estimate $V_{c_{max}}^{25}$ using empirical linear equations —$V_{c_{max}}^{25} = n1 \times LNC + n2$, with n1 and n2 being PFT-specific parameters. They are developed using the maximum carboxylation rate ($V_{c_{max}}$), maximum photosynthetic rate ($A_{max}$), and LNC records from the first version of the TRY database. EM2 is similar to EM1 except that it uses a smaller n1 parameter for EBF to consider an implicit phosphorus limitation. EM1 and EM2 suggest that fLNR are PFT-specific values.

*EM3 and EM4*[15]. The two models estimate $V_{c_{max}}^{25}$ using a power function of leaf nutrient content, with parameters in the function derived by fitting measurements from 24 studies. In particular, EM3 is $V_{c_{max}}^{25} = e^{3.712}LNC^{0.65}$ and EM4 is $V_{c_{max}}^{25} = e^{3.946}LNC^{[0.921 + 0.282\ln(LPC)]}LPC^{0.121}$. EM3 and EM4 suggest that fLNR decreases as LNC increases and EM4 indicates that fLNR decreases also as LPC increases.

*EM5*[17]. The EM5 model estimates $V_{c_{max}}^{25}$ based on PFT-specific environment-trait relationships, where the environment variables include precipitation, temperature, radiation, and CO$_2$. The environment-trait relationships were trained by multiple linear regressions of $V_{c_{max}}^{25}$ to climate, and the training dataset of $V_{c_{max}}^{25}$ was acquired from an early version of the TRY database supplemented by several studies.

*EO*[20]. The ecological optimality (EO) model estimates $V_{c_{max}}^{25}$ based on the assumption that plants minimize the carbon and water cost of photosynthesis while coordinating light-limited and RuBisCO-limited rates of photosynthesis to optimize leaf light use efficiency to maximize gross photosynthesis at the lowest enzymatic cost. It provides an analytic solution to dynamically estimate $V_{c_{max}}^{25}$ in models. The estimate of the EO model has been validated against the NS19 dataset[20] ($n = 3672$) at the growing season temperature. To facilitate the comparison in this study, we used a peaked Arrhenius temperature response function without temperature acclimation[62] to obtain $V_{c_{max}}^{25}$ from the EO model, while the original EO model estimate growing season $V_{c_{max}}^{25}$ using the temperature response function with temperature acclimation[62]. A minimum value of 0.08 for the temperature correction factor was applied to overcome extreme values[70].

*LUNA*[23]. The LUNA model dynamically adjusts the fraction of nitrogen invested in different components of photosynthetic and metabolic processes in leaves (i.e., light capture, electron transport, carboxylation, and respiration) and cell structures to maximize net photosynthesis (gross photosynthesis—photorespiration—dark respiration), under given environmental conditions and an optimal leaf nitrogen use strategy determined by the parameters of the LUNA model[22]. The driving environmental conditions include light, temperature, relative humidity, CO$_2$, and day length, while the parameters used in LUNA were estimated by fitting observed $V_{c_{max}}^{25}$ ($n = 833$)[14] to the LUNA model.

## Data availability statement

Most data used to support the findings of this study are publicly available. The NS19 dataset is a compilation of multiple datasets, some of which are not public according to their data policy. We encourage investigators to refer to the original studies cited in NS19 or contact N.G.S. for more information. The global leaf chlorophyll content map is available upon request to J.M.C. It is not public as the investigator is seeking the best practice to store and distribute this large volume of high-resolution remote sensing product. The three global leaf nitrogen maps are available at (1) EB17 at https://github.com/abhirupdatta/global_maps_of_plant_traits, (2) AMM18 at https://www.try-db.org/TryWeb/Data.php#59 and (3) CB20 at https://doi.org/10.6084/m9.figshare.11559852. The global maps of a fraction of leaf nitrogen invested in RuBisCO (fLNR) and $V_{c_{max}}^{25}$ are accessible via the Zenodo Data Repository (https://zenodo.org/record/5090497).

## Code availability statement

The code used to support the findings of this study is publicly available at www.github.com/lxzswr/Vcmax_fLNR. We used the Matlab package M_Map (https://www.eoas.ubc.ca/~rich/map.html) to create the maps in our study[71].

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

## Acknowledgements

X.L. and T.F.K. were supported by the NASA Terrestrial Ecology Program IDS Award NNH17AE86I. X.L. acknowledges support from the National University of Singapore. T.F.K. also acknowledges support by the Director, Office of Science, Office of Biological and

Environmental Research of the US Department of Energy under Contract DE-AC02-05CH11231 as part of the RUBISCO SFA. N.G.S. acknowledges support from Texas Tech University. C.X. acknowledges support from the Next Generation Ecosystem Experiment (NGEE) in the tropics, sponsored by the DOE Office of Science. H.C. was supported by the UK Research and Innovation (UKRI) Future Leaders Fellowship scheme [MR/T01993X/1]. I.C.P. has received funding from the European Research Council (ERC) under the European Union's Horizon 2020 research and innovation programme (grant agreement No: 787203 REALM). We thank the TRY database for making its data publicly available. We thank Dr. Ethan Butler for making their global leaf traits map publicly available. ORNL is managed by UT-Battelle, LLC, for the DOE under contract DE-AC05-1008 00OR22725. This manuscript has been co-authored by UT-Battelle, LLC under Contract No. DE-AC05-00OR22725 with the U.S. Department of Energy. The United States Government retains and the publisher, by accepting the article for publication, acknowledges that the United States Government retains a non-exclusive, paid-up, irrevocable, worldwide license to publish or reproduce the published form of this manuscript, or allow others to do so, for United States Government purposes. The Department of Energy will provide public access to these results of federally sponsored research in accordance with the DOE Public Access Plan (http://energy.gov/downloads/doe-public-access-plan).

## Author contributions

X.L. and T.F.K. conceived the idea and designed the study; X.L., T.F.K., J.M.C, I.C.P., H.C., N.G.S., A.P.W., and H.W. participated in the early-stage discussion. X.L. performed the analysis and led the writing; A.P.W., H.C., C.X. R.W., and Y.Z. contributed data, model outputs, and method suggestions; all co-authors contributed to the writing.

## Competing interests

The authors declare no competing interests.
