## [Peer Review File · Nature Communications]

REVIEWER COMMENTS

Reviewer #1 (Remarks to the Author):

This study presents a very thorough, interesting and relevant analysis of global variation in fLNR and Vcmax.

The results are highly relevant for global ecologists and modellers and are based on a very robust analysis.

I have no major critics on the study, as it is very well written, clearly presented and the analysis seems very robust.

However, I am missing one key thing in the paper. I am convinced that the global maps (especially the Vcmax map) will be widely used and cited by the global vegetation modelling community. I therefore recommend the authors to add a few sentences of “guidance” for modellers when using this map and it would be good to provide a “Vcmax uncertainty” map as well. Currently only fLNR uncertainty maps are provided. A map representing the uncertainty on the Vcmax map would be very helpful for modellers that want to extract a Vcmax value for a specific region out of the map...

The very high values for fLNR and Vcmax that you find in the Sahel, compare very well to recent observations in the Sahel by Sibret et al. 2021. I think this is a great illustration of the validity of the map, worth to mention in the discussion. For example, when referring to arid environments (line 282) and the relation with leaf nitrogen.

Related to that I found it a pity that no Savanna-type PFT was included in the PFT classification.

The authors present the 7 models as “7 competing hypotheses” (e.g. on line 240). I think this is not correct because M1& M2 and M3&M4 are actually based on the same hypothesis. So I think it is more correct to talk about 7 models representing 4 (or 5) hypotheses.

For the rest I only have a few minor comments and suggestions:

Line 114: the authors introduce leaf chlorophyll content as one of the 20 ‘environmental factors’. I would suggest to change the wording of that sentence, because the reader will expect mainly ‘external’ (abiotic) factors when you introduce them as such.

Line 116: the first ‘and’ should be a comma

Line 140: it is a bit odd/obvious to state that the correlation between RF fLNR and RF Vcmax was stronger than.... Because fLNR is directly derived from Vcmax observations...

Line 153: “tropical evergreen trees” ◊ I suggest to write “tropical evergreen forest” as I assume that you are referring here to variation between different tropical forests and not the variation between the trees within one single tropical forest site (which is probably very high, given the high species and functional diversity in these forests).

Line 312: when you mention equation 1 here, it is good to refer to the methods section

Line 313: alfa25 is mentioned for the first time here, but is only introduced properly on line 315.

Sibret, T., Verbruggen, W., Peaucelle, M., Verryckt, L. T., Bauters, M., Combe, M., ... Verbeeck, H. (2021). High photosynthetic capacity of Sahelian C3 and C4 plants. *Photosynthesis Research*.

Reviewer #2 (Remarks to the Author):

Overall the paper looks at the fraction of leaf nitrogen that is allocated to rubisco globally. The authors use a novel approach by combining several datasets, remote sensing products and models. Given the interest in recent years in remote sensing of leaf nitrogen, this is a timely paper. Especially since it has been debated how much canopy nitrogen actually contributes to photosynthesis.

While I very much appreciate the paper, and think it is a valuable contribution to the literature, there are some questions to the authors that need to be addressed.

Specific comments:

p. 3, line 67: The authors mention that empirical 'nutrient based' VCmax models use PFT specific linear equations to calculate VCmax from LNC, but how do they get LNC? From data? From RS, or an internal model result? This matters quite a bit, and data is not very much available.

p. 5, line 100: the references of 'remote sensing of leaf traits' are not the newest references, this field has been very active in the last years.

p. 5, line 104, where are the 8610 data sites of VCmax located? Since they are compiled from various data sources?

p. 5, line 106, why the use of the Butler map, rather than for instance the Moreno-Martinez et al. (2018, mentioned in the references, but not mentioned in the text of the paper) or the Boonman et al. (2020) map? These maps differ, and how does this influence the results of the paper? Also, the Butler map is given in N concentration, while the authors use N area based. How did the authors recalculate? Using SLA? And if this is the case, does this influence the relationship with LMA the authors find later in the paper?

In page 7, line 143-145 the authors do evaluate the VCmax models used, but I miss such an evaluation for the LNC input maps.

p. 7, line 138-139. Interesting to see that the fLNR in boreal and tropical zones is lower than in temperate zones, very unexpected given the N limitation in boreal and tropical zones, and the high N deposition in temperate zones. Can the authors explain this further?

p. 7, line 150-151. Was N-deposition also taken into account?

P. 10, line 185-187. EM5 captured the response of fLNR to leaf traits very well, but is not mentioned?

p. 10, line 189. EM5 is named as an example of an optimality model, but is the empirical climate model.

p. 11, line 197-199. Is the climate signal also related to N-deposition, as climate and Ndep are highly correlated? See also figure 3a, which resembles the Ndep map except for dry areas?

p. 11, line 202-203. How do I put these global average numbers into perspective?

p. 12, line 224. Where can I find these tropical forests? (see also p. 14, line 272-273)

p. 15, line 275-277. I think this statement is too general, as I mainly see this adjustment in forests and not in the other pft's. This is somewhat surprising, as many grasslands are also known to be P-limited. And needleleaf forests do not show this adjustment.

p. 15, line 285. Could there also be a link with fire? More fire > loss of N > stimulate plants to use

nitrogen more efficiently?

p. 16, line 302-304. The link of LNC and pH only holds within a certain range after which the soil gets too acid.

p. 17, line 332-333. The authors only look at the internal uncertainty of the LNC map, but not how this map compares to other more recent maps (see references above).

p. 18, line 347-248. While the optimal VCmax models performed well, EM5 also performed well (very similar to the optimal models). This should be discussed.

References:

Moreno-Martínez, Á. et al. A methodology to derive global maps of leaf traits using 753 remote sensing and climate data. *Remote Sens. Environ.* 218, 69–88 (2018)

Boonman et al. Assessing the reliability of predicted plant trait distributions at the global scale. *Global Ecology and Biogeography*. 29, 1034-1051 (2020)

Reviewer #3 (Remarks to the Author):

The received manuscript entitled “Global variation in the fraction of leaf nitrogen allocated to photosynthesis” aims to use a comprehensive database of in-situ observations and a novel remote sensing product of leaf chlorophyll, in combination with climate and soil characteristics, to examine the global distribution of fLNR using a random forest model (RF). Their estimates have been compared with some other parametric approaches. Moreover, the authors carried out different analyses to provide insights into the nitrogen-photosynthesis relationship of plants globally and an improved understanding of the global distribution of photosynthetic potential.

Although I appreciate the effort of the authors and I think it is a very interesting topic. I really see important flaws in the manuscript that need to be clearly addressed before publication. Especially because most of the insights that the authors are providing rely on the validity of the provided maps which, in my opinion, need further validation work.

1. Most of the traits maps provided in the literature used a much higher number of observations. Probably that is one of the reasons why no explicit maps of Vcmax have been provided so far. Because of this a much more detailed analysis of the results is needed.

*Ploton, P., Mortier, F., Réjou-Méchain, M., Barbier, N., Picard, N., Rossi, V., ... & Pélissier, R. (2020). Spatial validation reveals poor predictive performance of large-scale ecological mapping models. *Nature communications*, 11(1), 1-11.

2. Most of the papers dealing with trait estimations have paid special attention in how to find representative values of leaf traits at the considered scale. How did you deal with this issue?

*Appendix S8 in Butler, E. E. et al. Mapping local and global variability in plant trait distributions. *Proc. Natl. Acad. Sci.* 114, E10937–E10946 (2017).

*Moreno-Martínez, Á. et al. A methodology to derive global maps of leaf traits using remote sensing and

climate data. *Remote Sens. Environ.* 218, 69–88 (2018).

*Van Bodegom, P. M., Douma, J. C., & Verheijen, L. M. (2014). A fully traits-based approach to modeling global vegetation distribution. *Proceedings of the National Academy of Sciences*, 111(38), 13733-13738.

*Boonman, C. C., Benítez-López, A., Schipper, A. M., Thuiller, W., Anand, M., Cerabolini, B. E., ... & Santini, L. (2020). Assessing the reliability of predicted plant trait distributions at the global scale. *Global Ecology and Biogeography*, 29(6), 1034-1051.

3.This is also linked with my first concern. I think that your RF model could be overfitted. Have you tested if the convex hull of your input data is really representative of the whole planet where you are predicting?. Some box plots of your training data against the box plots of the rest of the planet should show that easily.

4.Have you compared the predicted maps with other sophisticated regression approaches (ANN, GPs, SVR...) to check if your results depend too much on the selected method (RF)? What about the accuracies, do you think that other methods could improve them?.

5.I also think that your uncertainty maps could be too optimistic. Unless you are completely sure that your model is not extrapolating, the variance of the RF model could be not reliable.

6.There are a variety of trait maps that could be also used in your work (including the Butler et al. ones). In fact, they present significant differences among them that could affect your results a lot. A comparison of the effect of them could be also very useful.

Check also: Boonman, C. C., Benítez-López, A., Schipper, A. M., Thuiller, W., Anand, M., Cerabolini, B. E., ... & Santini, L. (2020). Assessing the reliability of predicted plant trait distributions at the global scale. *Global Ecology and Biogeography*, 29(6), 1034-1051.

*Moreno-Martínez, Á. et al. A methodology to derive global maps of leaf traits using remote sensing and climate data. *Remote Sens. Environ.* 218, 69–88 (2018).

Global variation in the fraction of leaf nitrogen allocated to photosynthesis

NCOMMS-21-01819-T

Response to Reviewers

Authors: We appreciate the constructive comments from the reviewers and the invitation from the editor to submit a revised version. We have carefully followed the reviewers' suggestions to carry out additional analyses and improve our manuscript. Please see below our point-to-point responses in blue text following reviewer comments.

Reviewer #1 (Remarks to the Author):

R1C1: This study presents a very thorough, interesting and relevant analysis of global variation in fLNR and Vcmax.

The results are highly relevant for global ecologists and modellers and are based on a very robust analysis.

Authors: We appreciate the positive comments from the reviewer. We are excited about the perspective of providing a useful data-driven map of fLNR and Vcmax to improve the modelling of global vegetation dynamics.

R1C2: I have no major critics on the study, as it is very well written, clearly presented and the analysis seems very robust.

However, I am missing one key thing in the paper. I am convinced that the global maps (especially the Vcmax map) will be widely used and cited by the global vegetation modelling community. I therefore recommend the authors to add a few sentences of "guidance" for modellers when using this map and it would be good to provide a "Vcmax uncertainty" map as well. Currently only fLNR uncertainty maps are provided. A map representing the uncertainty on the Vcmax map would be very helpful for modellers that want to extract a Vcmax value for a specific region out of the map...

Authors: Thank you for the suggestion. Indeed, it was a missed opportunity to provide a guidance on the V_{cmax}^{25} map to facilitate the community use. We have added a paragraph in the discussion and a new figure (Fig. S8) to do so (L424 – L431).

"In addition, the data-driven V_{cmax}^{25} map provides a direct constraint on the spatial variations of vegetation photosynthetic capacity. We release the V_{cmax}^{25} map and its associated uncertainty

(i.e., one standard deviation of estimates from bagged trees in the RF) to facilitate large scale ecological and modelling studies (Fig. S8). Since the remote sensing retrieval of leaf chlorophyll content is for top leaves (Croft et al., 2020), and the phenological stages of trait samples are not often available, the $V_{c_{max}}^{25}$ estimated in our study can be interpreted as a multi-year average $V_{c_{max}}^{25}$ for top canopy leaves. The seasonality and within-canopy variations of $V_{c_{max}}^{25}$ are not accounted for in the map.”

Figure S8. The $V_{c_{max}}^{25}$ map estimated by the random forest model (RF). (a) $V_{c_{max}}^{25}$ and (b) the uncertainty of $V_{c_{max}}^{25}$, indicated by one standard deviation of the estimates from 200 bagged RF trees.

R1C3: The very high values for fLNR and Vcmax that you find in the Sahel, compare very well to recent observations in the Sahel by Sibret et al. 2021. I think this is a great illustration of the validity of the map, worth to mention in the discussion. For example, when referring to arid environments (line 282) and the relation with leaf nitrogen.

Related to that I found it a pity that no Savanna-type PFT was included in the PFT classification.

Authors: Thank for directing us to this new paper. We have added Sibret et al. 2021 in our discussion to support the high Vcmax25 and fLNR we found in the Sahel (L284-290).

“Several studies have reported that plants in arid environments (i.e., high VPD) tend to have a higher Amax and LNC (Sibret et al., 2021; Wright et al., 2003)...”

“Our results show that other than Amax and LNC, fLNR also increases with VPD, consistent with a recent study reporting higher nutrient use efficiency for plants in semi-arid ecosystems of the African Sahel (Sibret et al., 2021).”

In this study, we used the land cover map produced by the European Space Agency (ESA) CCI project. The ESA CCI land cover map does not have a savanna type PFT (see details here: http://maps.elie.ucl.ac.be/CCI/viewer/download/ESACCI-LC-Ph2-PUGv2_2.0.pdf). We used the ESA CCI map to parameterize the algorithm that derives leaf chlorophyll content from remote sensing (Croft et al., 2020). As leaf chlorophyll is the most important predictor in our study (Fig.

S5a), it is reasonable to use the same land cover map in this data-driven study. One study comparing MODIS and ESA CCI land cover suggested that “the Sahel and Savannah region...are characterized by approximately equal probabilities for multiple LC classes such as shrubland, grassland and bare/sparse vegetation” (Tsendbazar et al., 2017), therefore the fLNR and Vcmax of savannah should be similar to the values of grassland and shrubland.

R1C4: The authors present the 7 models as “7 competing hypotheses” (e.g. on line 240). I think this is not correct because M1& M2 and M3&M4 are actually based on the same hypothesis. So I think it is more correct to talk about 7 models representing 4 (or 5) hypotheses.

Authors: Thanks for pointing this out. We have revised the sentence to say “seven V_{cmax}^{25} models based on 5 competing hypotheses”.

For the rest I only have a few minor comments and suggestions:

R1C5: Line 114: the authors introduce leaf chlorophyll content as one of the 20 ‘environmental factors’. I would suggest to change the wording of that sentence, because the reader will expect mainly ‘external’ (abiotic) factors when you introduce them as such.

Authors: Motivated by your suggestion, we have revised the sentence to “We examined the relative importance of 20 environmental (i.e., biotic and abiotic) factors in estimating *in-situ* observations of V_{cmax}^{25} ...” to highlight the inclusion of both biotic and abiotic factors in our analysis.

We then listed the factors in the sequence of their relative importance identified by *the prediction of variable importance by permutation* analysis.

R1C6: Line 116: the first ‘and’ should be a comma

Authors: This has been corrected as suggested.

R1C7: Line 140: it is a bit odd/obvious to state that the correlation between RF fLNR and RF Vcmax was stronger than.... Because fLNR is directly derived from Vcamx observations...

Authors: We agree that this could cause confusion and have removed the statement.

R1C8: Line 153: “tropical evergreen trees” ◇ I suggest to write “tropical evergreen forest” as I assume that you are referring here to variation between different tropical forests and not the variation between the trees within one single tropical forest site (which is probably very high, given the high species and functional diversity in these forests).

Authors: This has been corrected as suggested.

R1C9: Line 312: when you mention equation 1 here, it is good to refer to the methods section
Authors: This has been corrected as suggested. We added “(see Methods)” after “Equation 1”

R1C10: Line 313: alfa25 is mentioned for the first time here, but is only introduced properly on line 315.

Authors: Thanks. We have updated the paragraph to define α^{25} and fNR upon their first appearances.

Sibret, T., Verbruggen, W., Peaucelle, M., Verryckt, L. T., Bauters, M., Combe, M., ... Verbeeck, H. (2021). High photosynthetic capacity of Sahelian C3 and C4 plants. Photosynthesis Research. <https://doi.org/10.1007/s11120-020-00801-3>

Reviewer #2 (Remarks to the Author):

R2C1: Overall the paper looks at the fraction of leaf nitrogen that is allocated to rubisco globally. The authors use a novel approach by combining several datasets, remote sensing products and models. Given the interest in recent years in remote sensing of leaf nitrogen, this is a timely paper. Especially since it has been debated how much canopy nitrogen actually contributes to photosynthesis.

While I very much appreciate the paper, and think it is a valuable contribution to the literature, there are some questions to the authors that need to be addressed.

Authors: We appreciate the constructive comments from the reviewer, and have taken them on board to improve the manuscript.

Specific comments:

R2C2: p. 3, line 67: The authors mention that empirical ‘nutrient based’ VCmax models use PFT specific linear equations to calculate VCmax from LNC, but how do they get LNC? From data? From RS, or an internal model result? This matters quite a bit, and data is not very much available.

Authors: We apologize for the confusion. We used a published global leaf nitrogen content (LNC; g/m²) map – EB17 (Butler et al., 2017) in our analysis, which is upscaled from in-situ observations using plant functional maps, climate and soil variables. The method for upscaling and validation of the LNC estimate are provided in EB17. We mentioned the use of this

published data in L106 in the Introduction and added a new paragraph in the Methods to introduce the LNC map (L457 – 470, please see our response to R1C5). The EB17 LNC data is publicly accessible at https://github.com/abhirupdatta/global_maps_of_plant_traits (accessed on Jun 2nd, 2021). We added this link in our data availability section.

Please kindly refer to our response to R1C5 where we describe the selection and use of LNC in detail.

R2C3: p. 5, line 100: the references of ‘remote sensing of leaf traits’ are not the newest references, this field has been very active in the last years.

Authors: We now cite more recent papers here per your suggestion (Asner et al., 2015; Croft et al., 2020; Moreno-Martínez et al., 2018; Serbin et al., 2015), and would be happy to add more if we missed any.

R2C4: p. 5, line 104, where are the 8610 data sites of VCmax located? Since they are compiled from various data sources?

Authors: Thank you. We should have pointed readers here to Fig. S6a, which provides the location data. We have updated the Introduction and the Methods to direct readers to the figure.

Figure S6a. The distribution of Vcmax25 observations. The size of the circles indicates the number of observations. The largest circle indicates ≥ 40 obs.

R2C5: p. 5, line 106, why the use of the Butler map, rather than for instance the Moreno-Martinez et al. (2018, mentioned in the references, but not mentioned in the text of the paper) or the Boonman et al. (2020) map? These maps differ, and how does this influence the results of the paper? Also, the Butler map is given in N concentration, while the authors use N area based. How did the authors recalculate? Using SLA? And if this is the case, does this influence the relationship with LMA the authors find later in the paper?

Authors: We apologize for the confusion. We have added a new section (L457-470) in the Methods to describe in details how we got LNC from EB17 (Butler et al., 2017). Additionally, motivated by the comment from the reviewer, we have also included two alternative leaf

nitrogen maps from AMM18 (Moreno-Martínez et al., 2018) and CB20 (Boonman et al., 2020) in our study.

“In our study, the LNC data we used to derive fLNR was acquired from EB17(Butler et al., 2017). EB17 provides data-driven estimates of global mass-based leaf nitrogen content (LNCm; unit: mg/g), specific leaf area (SLA; unit: m²/kg) and their associated uncertainties. We estimated LNC using the equation $LNC = LNCm/SLA$ (unit: g/m²). The uncertainty of LNC was propagated from the uncertainties of LNCm and SLA generated from 1000 bootstrapping tests. We note there are two alternative global LNC maps available: AMM18 (Moreno-Martínez et al., 2018) and CB20 (Boonman et al., 2020). Comparing the three LNC maps (Fig. S9), we found the spatial patterns of EB17 and AMM18 are similar to each other, with EB17 reported a relatively larger spatial gradient of LNC. CB20 LNC shows less evident spatial variation than EB17 and AMM18, and it has the lowest R² in validation among three products (Table S3). Since EB17 has the highest R² in validation among the three, and the weighting strategy of EB17 (i.e., unweighted by species abundance within a pixel) is close to the weighting strategy we adopted for $V_{C_{max}}^{25}$ estimation, we chose EB17 LNC map as the principal LNC in our analysis, but have also examined the impacts of using alternative LNC maps on our results (Fig. S10).”

Figure S9. Comparison of three published global leaf nitrogen content (LNC) maps (Boonman et al., 2020; Butler et al., 2017; Moreno-Martínez et al., 2018). (a,b,c) the spatial variation of LNC and (d,e,f) the spatial correlations between the LNC maps.

Table S3. Evaluation of LNCm and SLA of the three products: EB17(Butler et al., 2017), AMM18(Moreno-Martínez et al., 2018), CB20(Boonman et al., 2020)

Products	LNCm (mass-based; mg/g)		SLA (m ² /kg)	
	R ² or pseudo R ²	RMSE or RMPSE	R ² or pseudo R ²	RMSE or RMPSE
EB17	0.548	6.18	0.602	6.13
AMM18	0.539	2.30	0.582	3.19
CB20	< 0.3		< 0.2	

We also added a new section in the Discussion to examine whether the choice of leaf nitrogen maps would influence our results (L359-373).

“The choice of LNC map is another source of uncertainty in the derivation of fLNR. There are several global LNC maps available other than the EB17 (Butler et al., 2017) map we used, namely AMM18 (Moreno-Martínez et al., 2018) and CB20 (Boonman et al., 2020). Each product has been validated in their respective studies (Table S3). To examine the uncertainty incurred by the choice of LNC maps, we calculated fLNR using each of the three LNC maps. The three resulting fLNR maps show similar spatial patterns (Fig. S10), with the spatial correlation coefficients (r) between them ranging from 0.57 to 0.71 ($p < 0.01$). Examining the sensitivities of fLNR to environmental variables, we found the fLNR based on EB17 and AMM18 demonstrated similar sensitivities – fLNR is most sensitive to LMA, LPC, VPD, PAR and soil pH. Meanwhile, the fLNR based on CB20 is most sensitive to soil pH, LMA, VPD, air temperature, and soil sand percentage. Noting that CB20 LNC map has lower R² in its cross-validation compared to that of EB17 and AMM18 (Table S3), we have more confidence in the fLNR maps based on EB17 and AMM18. In our study, we used EB17 as the principal LNC map as it demonstrated the highest R² in validation (Table S3), and the weighting strategy of EB17 (i.e., unweighted by species abundance within a pixel) is close to the weighting strategy we adopted for $V_{c_{max}}^{25}$.”

Figure S10. Three global fLNR maps estimated using alternative leaf nitrogen content maps (Boonman et al., 2020; Butler et al., 2017; Moreno-Martínez et al., 2018). (a,b,c) the spatial variation of fLNR, (d,e,f) the spatial correlations between the fLNR maps and (g,h,i) the sensitivities of fLNR to environmental factors.

The calculation of fLNR from LNC (which is LNC_m/SLA , or equal to $LNC_m \cdot LMA$) has limited influence on the sensitivity of fLNR to LMA. The reason is that for fLNR sensitivity analysis, we examined the additive effect of LMA on fLNR, while for the calculation of fLNR, LMA is essentially in the denominator of the equation. Therefore, most of the spatial variation in fLNR originates from the variations of the numerator - the V_{cmax}^{25} (which was derived from remotely sensed leaf chlorophyll content, plant functional types, precipitation and soil pH not LMA). This is also seen in the stronger spatial correlations between fLNR maps (Fig. S10d,e,f) than the spatial correlations between the LNC maps (Fig. S9d,e,f).

R2C6: In page 7, line 143-145 the authors do evaluate the V_{cmax} models used, but I miss such an evaluation for the LNC input maps.

Authors: We used published global LNC maps, which have examined their products in their respective studies. We have included their validation results in Table S3 (please see our response to R2C5).

R2C7: p. 7, line 138-139. Interesting to see that the fLNR in boreal and tropical zones is lower than in temperate zones, very unexpected given the N limitation in boreal and tropical zones, and the high N deposition in temperate zones. Can the authors explain this further?

Authors: That is an interesting point and we are glad to discuss more.

In this study, we explore the spatial sensitivities of fLNR to various biotic and abiotic factors. First, we did not see a strong response of fLNR to changes in soil N over space (Fig. 3b). While we acknowledge that some studies have suggested soil N addition/deposition influenced photosynthesis, the significance of the N deposition on photosynthesis is dependent on biome types, N deposition load (Fleischer et al., 2013), and time after the deposition (Liang et al., 2020). As we focus on the spatial variation of fLNR in this study, the localized and time-dependent changes can be minimized in the average global pattern we see.

Second, it has been recently suggested that leaf photosynthetic capacity is decoupled from soil N supply, as N supply is generally enough to support leaf-level photosynthesis (Peng et al., 2021). The changes in leaf photosynthesis are likely incurred by increasing CO₂ or acclimation to local climate. Please also note that the discussed influences of N deposition are mostly for LNC and photosynthetic capacities, we are not sure there are studies showing that N deposition can influence fLNR.

Third, it is very likely the influence of N deposition on soil N, if any, has already been included in the soil N map we used. Soilgrids use ground measured soil profiles from WoSIS database for its global upscaling. Among the soil profiles from WoSIS, at least 47.7% (maximum 87.7%) samples are collected between 1980 and 2019 (Batjes et al., 2020). We would expect the strong N deposition during this period (Zhang et al., 2017) has been implicitly incorporated into the soil N map, and our study found no evident soil N impacts on fLNR (Fig. 3).

We have added the discussion above on nitrogen deposition (L316-325) in the manuscript.

At last, while we are unsure that fLNR should be indicative of soil N availability, our study suggests that the lower fLNR in boreal and tropical ecosystems are due to the coordination of leaf traits along the leaf economic spectrum (Fig. 3b-c). With heavier investment of nutrients in the structural build-up of evergreen leaves, less nitrogen is available for photosynthesis (i.e., lower fLNR) in some tropical and boreal regions.

R2C8: p. 7, line 150-151. Was N-deposition also taken into account?

Authors: Please see our response to R2C7.

R2C9: P. 10, line 185-187. EM5 captured the response of fLNR to leaf traits very well, but is not mentioned?

Authors: We have modified the statement to “the optimal V_{cmax}^{25} models (i.e., EO and LUNA) and EM5 captured the response of fLNR to leaf traits PC1 well...” following the reviewer’s suggestion.

R2C10: p. 10, line 189. EM5 is named as an example of an optimality model, but is the empirical climate model.

Authors: We apologize for this and have corrected it.

R2C11: p. 11, line 197-199. Is the climate signal also related to N-deposition, as climate and Ndep are highly correlated? See also figure 3a, which resembles the Ndep map except for dry areas?

Authors: Please see our response to R2C7. Our results do not show that soil N (which included N deposition signals) has evident influence on fLNR.

R2C12: p. 11, line 202-203. How do I put these global average numbers into perspective?

Authors: The numbers we reported here indicate the overall quantitative effect of each type of factor (i.e., leaf traits, climate and soil) on the spatial variation of fLNR. It helps us understand, in a relative sense, which process governs the spatial variation of fLNR.

In the section following the statement, we disassembled these numbers into the sensitivities of fLNR to individual biotic and abiotic factors. For example, we disassembled the total effect from “climate” on fLNR into the effects of “temperature”, “precipitation”, “PAR”, “VPD” and “SWC” on fLNR (Fig. 3b).

R2C13: p. 12, line 224. Where can I find these tropical forests? (see also p. 14, line 272-273)

Authors: We examined the spatial sensitivities of fLNR to environment variables. Therefore, the places that have lower leaf phosphorus content (LPC) are likely to have a stronger LPC limitation effect on fLNR (Fig. R2), e.g., western Congo basin forests and central Amazon forests.

Figure R2. The leaf phosphorus content (LPC) of evergreen broadleaf forests in the tropics. Note that the range of tropical LPC (0 to 0.12 g/m²) is much lower than the global range (0 to 0.40 g/m²).

R2C14: p. 15, line 275-277. I think this statement is too general, as I mainly see this adjustment in forests and not in the other pft's. This is somewhat surprising, as many grasslands are also known to be P-limited. And needleleaf forests do not show this adjustment.

Authors: In our study, we found the fLNR of EBF and MF have strong positive sensitivity to LPC, while other plant functional types (including grasslands and ENF) only show limited LPC impact on fLNR. Following the comment, we have added a statement in the discussion to acknowledge the different impact of LPC on fLNR across plant functional types (L275-280):

“In addition, we note that the productivity of some grasslands (Dong et al., 2019; Fay et al., 2015) and boreal forests (Braun et al., 2010; Giesler et al., 2002) has also been reported to be limited by phosphorus availability, however, we did not detect strong positive dependence of fLNR on LPC globally for these ecosystems in our study. The difference potentially suggests that the phosphorus limitation of grasslands and boreal forests is not as prevalent as that for tropical forests and mixed forests (though some mixed forests are in the boreal region)”.

R2C15: p. 15, line 285. Could there also be a link with fire? More fire > loss of N > stimulate plants to use nitrogen more efficiently?

Authors: We feel the fire – N availability– plants link is outside of the scope of our study and prefer to refrain from speculative discussion here (e.g., low intensity fire can increase nitrogen availability in some studies (Schoch and Binkley, 1986)). Other than the reasons we provided in the response to R2C7 that soil N has no evident impacts on fLNR, we are unsure whether fire, often sporadic in nature can shape soil N supply at the global scale.

R2C16: p. 16, line 302-304. The link of LNC and pH only holds within a certain range after which the soil gets too acid.

Authors: Thank you for this point. We only examined the impact of pH on fLNR, not LNC, in our study (Figure 3). We would be happy to check out the paper if this is a study we missed.

R2C17: p. 17, line 332-333. The authors only look at the internal uncertainty of the LNC map, but not how this map compares to other more recent maps (see references above).

Authors: Thank you. We fully agree and have included two more LNC datasets (Boonman et al., 2020; Moreno-Martínez et al., 2018) in our analysis. Please see our response to R2C5.

R2C18: p. 18, line 347-248. While the optimal VCmax models performed well, EM5 also performed well (very similar to the optimal models). This should be discussed.

Authors: We agree that EM5 performed well in showing a large spatial variation of Vcmax25 and capturing the influence of other leaf traits on fLNR. We have added this statement in our results (please see our response to R2C9). However, EM5 did not well capture the impacts of climate and soil (Fig. 2) and thus results in very different spatial patterns of Vcmax25 and fLNR than those from the optimality model and the RF (Fig. S1 and S2).

References:

Moreno-Martínez, Á. et al. A methodology to derive global maps of leaf traits using 753 remote sensing and climate data. *Remote Sens. Environ.* 218, 69–88 (2018)

Boonman et al. Assessing the reliability of predicted plant trait distributions at the global scale. *Global Ecology and Biogeography.* 29, 1034-1051 (2020)

Reviewer #3 (Remarks to the Author):

R3C1: The received manuscript entitled “Global variation in the fraction of leaf nitrogen allocated to photosynthesis” aims to use a comprehensive database of in-situ observations and a novel remote sensing product of leaf chlorophyll, in combination with climate and soil characteristics, to examine the global distribution of fLNR using a random forest model (RF). Their estimates have been compared with some other parametric approaches. Moreover, the authors carried out different analyses to provide insights into the nitrogen-photosynthesis relationship of plants globally and an improved understanding of the global distribution of photosynthetic potential.

Authors: Thank you for reviewing our paper and for your very useful suggestions.

R3C2: Although I appreciate the effort of the authors and I think it is a very interesting topic. I really see important flaws in the manuscript that need to be clearly addressed before

publication. Especially because most of the insights that the authors are providing rely on the validity of the provided maps which, in my opinion, need further validation work.

Authors: We appreciate the constructive comments. We have used your comments to improve our study, including extensive additional validation efforts. Please see details below.

R3C3: 1. Most of the traits maps provided in the literature used a much higher number of observations. Probably that is one of the reasons why no explicit maps of V_{cmax} have been provided so far. Because of this a much more detailed analysis of the results is needed.

*Ploton, P., Mortier, F., Réjou-Méchain, M., Barbier, N., Picard, N., Rossi, V., ... & Pélissier, R. (2020). Spatial validation reveals poor predictive performance of large-scale ecological mapping models. *Nature communications*, 11(1), 1-11.

Authors: We agree with the reviewer that the production of a data-driven V_{cmax}^{25} map has been challenging. In recent years, with the deluge of public trait data and the new remote sensing data, we are reaching a point of having enough information to infer global V_{cmax}^{25} . We are confident in this because 1) we have amassed a considerable amount of V_{cmax}^{25} observations ($n = 8610$) from multiple sources and 2) we have a new dataset of global leaf chlorophyll content, which is known to relate to V_{cmax}^{25} (Croft et al., 2017). We appreciate the sentiment to carry out more detailed examination of the V_{cmax}^{25} map, however, and have added more validation analyses in this revised version:

1. We added a spatial cross-validation. In addition to the out-of-the-bag (OOB) test we did in the original manuscript, we conducted a spatial cross-validation to validate the V_{cmax}^{25} map, following the paper recommended by the reviewer (Ploton et al., 2020). We removed the validation data points within 1.5 degree (~ 150 km) of the training samples to eradicate the influence of spatial autocorrelation on extrapolation.

We added the following statement in the Methods (L522-532)

“We trained the RF using the top 4 predictors and used both conventional and spatial cross-validation to examine the reliability of the RF. We used 80% of samples for training and 20% for validation. The spatial cross-validation is similar to conventional cross-validation, but we removed the validation data points within 1.5 degree (~ 150 km) of the training samples to avoid spatial autocorrelation (Ploton et al., 2020). The spatial cross-validation showed an $R^2 = 0.52 \pm 0.36$ and an RMSE of $39.9 \pm 14.5 \mu\text{mol m}^{-2} \text{s}^{-1}$, while the conventional cross-validation showed an $R^2 = 0.52 \pm 0.01$ and an RMSE of $20.7 \pm 0.3 \mu\text{mol m}^{-2} \text{s}^{-1}$ (Fig. S11). The accuracy of

the conventional cross-validation is comparable to previous trait upscaling studies (Boonman et al., 2020; Butler et al., 2017; Moreno-Martínez et al., 2018; Van Bodegom et al., 2014)”

Figure S11. The validation of the RF $V_{C_{max}}^{25}$. ‘C.V.’ indicates conventional cross-validation, ‘S.C.V.’ indicates spatial cross-validation, ‘All’ indicates using all available samples for validation. (a) R^2 of validation; (b) RMSE of validation. Error bars indicate one standard deviation of validation runs.

2. We examined the representatives of samples for global extrapolation. To examine whether the current sample has a good coverage of the globe, we assessed the histogram of predictors for training samples and the globe. We found that our training samples cover the global range of all predictors well (Fig. S12).

Figure S12. The representativeness of samples for global extrapolation.

3. We added a new section to describe the ecological meaning of the $V_{\max 25}$ estimated and its associated uncertainty (Please see our response to R1C2)

R3C4: 2. Most of the papers dealing with trait estimations have paid special attention in how to find representative values of leaf traits at the considered scale. How did you deal with this issue?

*Appendix S8 in Butler, E. E. et al. Mapping local and global variability in plant trait distributions. *Proc. Natl. Acad. Sci.* 114, E10937–E10946 (2017).

*Moreno-Martínez, Á. et al. A methodology to derive global maps of leaf traits using remote sensing and climate data. *Remote Sens. Environ.* 218, 69–88 (2018).

*Van Bodegom, P. M., Douma, J. C., & Verheijen, L. M. (2014). A fully traits-based approach to modeling global vegetation distribution. *Proceedings of the National Academy of Sciences*, 111(38), 13733-13738.

*Boonman, C. C., Benítez-López, A., Schipper, A. M., Thuiller, W., Anand, M., Cerabolini, B. E., ... & Santini, L. (2020). Assessing the reliability of predicted plant trait distributions at the global scale. *Global Ecology and Biogeography*, 29(6), 1034-1051.

Authors: Thanks for the question. We fully agree that finding representative values of V_{cmax25} of each pixel (community level trait value) is challenging, as we know V_{cmax25} within an ecosystem not only changes with species/genus, but also change with seasons and location of the leaves in the canopy. However, this high-level heterogeneity information is often lacking in trait databases.

We have indeed tested using species and genus abundance weighted trait values to train the RF for V_{cmax25} estimation, however, several issues arose in the process: 1) the abundance weights for V_{cmax25} are not comparable with those of key predictors/covariates (e.g. LNC, LMA and LPC), because they have different number of observations across the globe; 2) using community weighted average values leads to fewer samples for training (from $n = 8610$ to $n = 429$ as we aggregated these trait values to the community level) and therefore makes the RF more susceptible to overfitting.

To address these challenges, we used all traits in the random forest (RF) model without specifying their weights. We expect the RF can identify the general gradient in samples by using a large number of samples to help trees in the RF to make the right assignments. We understand such a strategy generates uncertainties as the estimates vary between runs when different chunks of data are sampled, but it also allows us to quantify the uncertainty of V_{cmax25} through running a large ensemble of trees. We added a statement to acknowledge this in our method section (L490-495):

“We did not apply species/genus abundance weights to trait values for training and validation purposes, because 1) the available species abundance information at the community level for V_{cmax}^{25} was not the same as that for other leaf traits (i.e., leaf nitrogen content) that are potential predictors; 2) we had much smaller number of samples for training if we aggregated trait values to the community-level (from $n = 8610$ to $n = 429$), which led to a greater risk of overfitting the RF model.”

R3C5: 3. This is also linked with my first concern. I think that your RF model could be overfitted. Have you tested if the convex hull of your input data is really representative of the whole planet where you are predicting?. Some box plots of your training data against the box plots of the rest of the planet should show that easily.

Authors: Please see our response to R3C3 point 2 where we show our input data for training the RF is representative of the global values.

R3C6: 4. Have you compared the predicted maps with other sophisticated regression approaches (ANN, GPs, SVR...) to check if your results depend too much on the selected method (RF)? What about the accuracies, do you think that other methods could improve them?.

Authors: Thanks for mentioning that. Yes, we used ANN in the early stage of our study (Fig. R3). We were concerned that ANN was subject to serious overfitting as its performance was sensitive to the selection of variables and the number of nodes and layers of the neural network. Though the mean values of V_{cmax25} from ANN estimates were similar to that of RF (Fig. R3 vs Fig. S8), the uncertainty of V_{cmax25} from ANN was substantially larger due to overfitting.

Figure R3. The V_{cmax}^{25} map estimated by Artificial Neural Network (ANN) and its associated uncertainties. First reported in 2019 American Geophysical Union general meeting (<https://agu.confex.com/agu/fm19/meetingapp.cgi/Paper/545500>).

As the study advanced, we decided to use RF for two reasons: 1) RF is good at dealing with categorical variables (James et al., 2013; Moreno-Martínez et al., 2018) and for some key inputs such as plant functional types and koeppen climate zones, we can directly use these variables for training while methods like ANN have to convert them to continuous variables or divide

samples by categories; 2) the tree-based RF method is compatible with *the prediction of variable importance by permutation*, a widely used method to select key variables for machine learning based on their importance (e.g., Terrer et al., 2019). Using a reduced number of variables can help avoid overfitting, especially when the number of candidate predictors in our study is not small (n=20) compared to the number of samples (n = 8610). We added a new section in the manuscript (L485-490) based on the statement above to justify our choice of the RF.

As several recent studies have suggested that RF is a robust method for the extrapolation of leaf traits (Boonman et al., 2020; Moreno-Martínez et al., 2018), we did not intend to examine a whole set of machine learning methods here. One comparison study has suggested that RF performance is as good as general linear models (GLM) and general additive models (GAM) if not better than them (Boonman et al., 2020).

R3C7: 5.I also think that your uncertainty maps could be too optimistic. Unless you are completely sure that your model is not extrapolating, the variance of the RF model could be not reliable.

Authors: Thank you for the comment. Please see our response to R3C3 on cross-validation and the representativeness of samples, which hopefully can alleviate your concerns on the uncertainty of extrapolation.

In addition to the reasons above, the new remote sensing leaf chlorophyll content (Chl) dataset provides a novel and direct constraint on the spatial variation of leaf photosynthetic capacity (Croft et al., 2017; Luo et al., 2019). Our importance analysis suggested that Chl is the most important predictor in our random forest model (Fig. S5a). The use of this independent and observational Chl map can help us reduce the uncertainty incurred from extrapolation.

R3C8: 6. There are a variety of trait maps that could be also used in your work (including the Butler et al. ones). In fact, they present significant differences among them that could affect your results a lot. A comparison of the effect of them could be also very useful.

Check also: Boonman, C. C., Benítez-López, A., Schipper, A. M., Thuiller, W., Anand, M., Cerabolini, B. E., ... & Santini, L. (2020). Assessing the reliability of predicted plant trait distributions at the global scale. *Global Ecology and Biogeography*, 29(6), 1034-1051.

*Moreno-Martínez, Á. et al. A methodology to derive global maps of leaf traits using remote sensing and climate data. *Remote Sens. Environ.* 218, 69–88 (2018).

Authors: Thank you for bringing this up. We fully agree that the uncertainty incurred by the use of different LNC dataset may influence our results. We have therefore added new analysis and a

new section in the Discussion to examine whether the choice of leaf nitrogen maps would influence our results (L359-373).

“The choice of LNC map is another source of uncertainty in the derivation of fLNR. There are several global LNC maps available other than the EB17(Butler et al., 2017) we used, namely AMM18(Moreno-Martínez et al., 2018) and CB20(Boonman et al., 2020). Each product has been validated in their respective studies (Table S3). To examine the uncertainty incurred by the choice of LNC maps, we calculated fLNR using each of the three LNC maps. The three resultant fLNR maps show similar spatial patterns (Fig. S10), with the spatial correlation coefficients (r) between them range from 0.57 to 0.71 ($p < 0.01$). Examining the sensitivities of fLNR to environmental variables, we found the fLNR based on EB17 and AMM18 demonstrated similar sensitivities – fLNR is most sensitive to LMA, LPC, VPD, PAR and soil pH. Meanwhile, the fLNR based on CB20 is most sensitive to soil pH, LMA, VPD, air temperature and soil sand percentage. Noting that CB20 LNC map has lower R^2 in its cross-validation compared to that of EB17 and AMM18 (Table S3), we have more confidence in the fLNR maps based on EB17 and AMM18. In our study, we used EB17 as the principal LNC map as it demonstrated the highest R^2 in validation.”

Figure S10. Three global fLNR maps estimated using alternative leaf nitrogen content maps (Boonman et al., 2020; Butler et al., 2017; Moreno-Martínez et al., 2018). (a,b,c) the spatial variation of fLNR, (d,e,f) the spatial correlations between the fLNR maps and (g,h,i) the sensitivities of fLNR to environmental factors.

Table S3. Evaluation of LNCm and SLA of the three products: EB17(Butler et al., 2017), AMM18(Moreno-Martínez et al., 2018), CB20(Boonman et al., 2020)

Products	LNCm (mass-based; mg/g)		SLA (m ² /kg)	
	R ² or pseudo R ²	RMSE or RMPSE	R ² or pseudo R ²	RMSE or RMPSE
EB17	0.548	6.18	0.602	6.13
AMM18	0.539	2.30	0.582	3.19
CB20	< 0.3		< 0.2	

Asner, G.P., Martin, R.E., Anderson, C.B., Knapp, D.E., 2015. Quantifying forest canopy traits: Imaging spectroscopy versus field survey. *Remote Sens. Environ.* 158, 15–27.

<https://doi.org/10.1016/j.rse.2014.11.011>

Batjes, N.H., Ribeiro, E., Van Oostrum, A., 2020. Standardised soil profile data to support global mapping and modelling (WoSIS snapshot 2019). *Earth Syst. Sci. Data* 12, 299–320.

<https://doi.org/10.5194/essd-12-299-2020>

Boonman, C.C.F., Benítez-López, A., Schipper, A.M., Thuiller, W., Anand, M., Cerabolini, B.E.L., Cornelissen, J.H.C., Gonzalez-Melo, A., Hattingh, W.N., Higuchi, P., Laughlin, D.C., Onipchenko, V.G., Peñuelas, J., Poorter, L., Soudzilovskaia, N.A., Huijbregts, M.A.J., Santini, L., 2020. Assessing the reliability of predicted plant trait distributions at the global scale. *Glob. Ecol. Biogeogr.* 1–18. <https://doi.org/10.1111/geb.13086>

Braun, S., Thomas, V.F.D., Quiring, R., Flückiger, W., 2010. Does nitrogen deposition increase forest production? The role of phosphorus. *Environ. Pollut.* 158, 2043–2052.

<https://doi.org/10.1016/j.envpol.2009.11.030>

Butler, E.E., Datta, A., Flores-Moreno, H., Chen, M., Wythers, K.R., Fazayeli, F., Banerjee, A., Atkin, O.K., Kattge, J., Amiaud, B., Blonder, B., Boenisch, G., Bond-Lamberty, B., Brown, K.A., Byun, C., Campetella, G., Cerabolini, B.E.L., Cornelissen, J.H.C., Craine, J.M., Craven, D., de Vries, F.T., Díaz, S., Domingues, T.F., Forey, E., González-Melo, A., Gross, N., Han, W., Hattingh, W.N., Hickler, T., Jansen, S., Kramer, K., Kraft, N.J.B., Kurokawa, H., Laughlin, D.C., Meir, P., Minden, V., Niinemets, Ü., Onoda, Y., Peñuelas, J., Read, Q., Sack, L., Schamp, B., Soudzilovskaia, N.A., Spasojevic, M.J., Sosinski, E., Thornton, P.E., Valladares, F., van Bodegom, P.M., Williams, M., Wirth, C., Reich, P.B., 2017. Mapping local and global variability in plant trait distributions. *Proc. Natl. Acad. Sci.* 114, E10937–E10946.

<https://doi.org/10.1073/pnas.1708984114>

Croft, H., Chen, J.M., Luo, X., Bartlett, P., Chen, B., Staebler, R.M., 2017. Leaf chlorophyll content as a proxy for leaf photosynthetic capacity. *Glob. Chang. Biol.* 23, 3513–3524.

<https://doi.org/10.1111/gcb.13599>

- Croft, H., Chen, J.M., Wang, R., Mo, G., Luo, S., Luo, X., He, L., Gonsamo, A., Arabian, J., Zhang, Y., Simic-Milas, A., Noland, T.L., He, Y., Homolová, L., Malenovský, Z., Yi, Q., Beringer, J., Amiri, R., Hutley, L., Arellano, P., Stahl, C., Bonal, D., 2020. The global distribution of leaf chlorophyll content. *Remote Sens. Environ.* 236. <https://doi.org/10.1016/j.rse.2019.111479>
- Dong, C., Wang, W., Liu, H., Xu, X., Zeng, H., 2019. Temperate grassland shifted from nitrogen to phosphorus limitation induced by degradation and nitrogen deposition: Evidence from soil extracellular enzyme stoichiometry. *Ecol. Indic.* 101, 453–464. <https://doi.org/https://doi.org/10.1016/j.ecolind.2019.01.046>
- Fay, P.A., Prober, S.M., Harpole, W.S., Knops, J.M.H., Bakker, J.D., Borer, E.T., Lind, E.M., MacDougall, A.S., Seabloom, E.W., Wragg, P.D., Adler, P.B., Blumenthal, D.M., Buckley, Y.M., Chu, C., Cleland, E.E., Collins, S.L., Davies, K.F., Du, G., Feng, X., Firn, J., Gruner, D.S., Hagenah, N., Hautier, Y., Heckman, R.W., Jin, V.L., Kirkman, K.P., Klein, J., Ladwig, L.M., Li, Q., McCulley, R.L., Melbourne, B.A., Mitchell, C.E., Moore, J.L., Morgan, J.W., Risch, A.C., Schütz, M., Stevens, C.J., Wedin, D.A., Yang, L.H., 2015. Grassland productivity limited by multiple nutrients. *Nat. Plants* 1, 1–5. <https://doi.org/10.1038/nplants.2015.80>
- Fleischer, K., Rebel, K.T., Van Der Molen, M.K., Erisman, J.W., Wassen, M.J., Van Loon, E.E., Montagnani, L., Gough, C.M., Herbst, M., Janssens, I.A., Gianelle, D., Dolman, A.J., 2013. The contribution of nitrogen deposition to the photosynthetic capacity of forests. *Global Biogeochem. Cycles* 27, 187–199. <https://doi.org/10.1002/gbc.20026>
- Giesler, R., Petersson, T., Högberg, P., 2002. Phosphorus limitation in boreal forests: Effects of aluminum and iron accumulation in the humus layer. *Ecosystems* 5, 300–314. <https://doi.org/10.1007/s10021-001-0073-5>
- James, G., Witten, D., Hastie, T., Tibishirani, R., 2013. *An Introduction to Statistical Learning with Applications in R (older version)*, Springer Texts in Statistics. New York : Springer, [2013] ©2013.
- Liang, X., Zhang, T., Lu, X., Ellsworth, D.S., BassiriRad, H., You, C., Wang, D., He, P., Deng, Q., Liu, H., Mo, J., Ye, Q., 2020. Global response patterns of plant photosynthesis to nitrogen addition: A meta-analysis. *Glob. Chang. Biol.* 26, 3585–3600. <https://doi.org/10.1111/gcb.15071>
- Luo, X., Croft, H., Chen, J.M., He, L., Keenan, T.F., 2019. Improved estimates of global terrestrial photosynthesis using information on leaf chlorophyll content. *Glob. Chang. Biol.* 25, 2499–2514. <https://doi.org/10.1111/gcb.14624>
- Moreno-Martínez, Á., Camps-Valls, G., Kattge, J., Robinson, N., Reichstein, M., van Bodegom, P., Kramer, K., Cornelissen, J.H.C., Reich, P., Bahn, M., Niinemets, Ü., Peñuelas, J., Craine, J.M., Cerabolini, B.E.L., Minden, V., Laughlin, D.C., Sack, L., Allred, B., Baraloto, C., Byun, C., Soudzilovskaia, N.A., Running, S.W., 2018. A methodology to derive global maps of leaf traits using remote sensing and climate data. *Remote Sens. Environ.* 218, 69–88. <https://doi.org/10.1016/j.rse.2018.09.006>
- Peng, Y., Bloomfield, K.J., Cernusak, L.A., Domingues, T.F., Colin Prentice, I., 2021. Global climate and nutrient controls of photosynthetic capacity. *Commun. Biol.* 4, 1–9. <https://doi.org/10.1038/s42003-021-01985-7>
- Ploton, P., Mortier, F., Réjou-Méchain, M., Barbier, N., Picard, N., Rossi, V., Dormann, C., Cornu, G., Viennois, G., Bayol, N., Lyapustin, A., Gourlet-Fleury, S., Pélissier, R., 2020. Spatial

- validation reveals poor predictive performance of large-scale ecological mapping models. *Nat. Commun.* 11, 1–11. <https://doi.org/10.1038/s41467-020-18321-y>
- Schoch, P., Binkley, D., 1986. Prescribed burning increased nitrogen availability in a mature loblolly pine stand. *For. Ecol. Manage.* 14, 13–22. [https://doi.org/https://doi.org/10.1016/0378-1127\(86\)90049-6](https://doi.org/https://doi.org/10.1016/0378-1127(86)90049-6)
- Serbin, S.P., Singh, A., Desai, A.R., Dubois, S.G., Jablonski, A.D., Kingdon, C.C., Kruger, E.L., Townsend, P.A., 2015. Remotely estimating photosynthetic capacity, and its response to temperature, in vegetation canopies using imaging spectroscopy. *Remote Sens. Environ.* 167, 78–87. <https://doi.org/10.1016/j.rse.2015.05.024>
- Sibret, T., Verbruggen, W., Peaucelle, M., Verryckt, L.T., Bauters, M., Combe, M., Boeckx, P., Verbeeck, H., 2021. High photosynthetic capacity of Sahelian C3 and C4 plants. *Photosynth. Res.* 147, 161–175. <https://doi.org/10.1007/s11120-020-00801-3>
- Terrer, C., Jackson, R.B., Prentice, I.C., Keenan, T.F., Kaiser, C., Vicca, S., Fisher, J.B., Reich, P.B., Stocker, B.D., Hungate, B.A., Peñuelas, J., McCallum, I., Soudzilovskaia, N.A., Cernusak, L.A., Talhelm, A.F., Van Sundert, K., Piao, S., Newton, P.C.D., Hovenden, M.J., Blumenthal, D.M., Liu, Y.Y., Müller, C., Winter, K., Field, C.B., Viechtbauer, W., Van Lissa, C.J., Hoosbeek, M.R., Watanabe, M., Koike, T., Leshyk, V.O., Polley, H.W., Franklin, O., 2019. Nitrogen and phosphorus constrain the CO₂ fertilization of global plant biomass. *Nat. Clim. Chang.* 9, 684–689. <https://doi.org/10.1038/s41558-019-0545-2>
- Tsendbazar, N.E., de Bruin, S., Herold, M., 2017. Integrating global land cover datasets for deriving user-specific maps. *Int. J. Digit. Earth* 10, 219–237. <https://doi.org/10.1080/17538947.2016.1217942>
- Van Bodegom, P.M., Douma, J.C., Verheijen, L.M., 2014. A fully traits-based approach to modeling global vegetation distribution. *Proc. Natl. Acad. Sci. U. S. A.* 111, 13733–13738. <https://doi.org/10.1073/pnas.1304551110>
- Wright, I.J., Reich, P.B., Westoby, M., 2003. Least-Cost Input Mixtures of Water and Nitrogen for Photosynthesis 161.
- Zhang, B., Tian, H., Lu, C., Dangal, S., Yang, J., Pan, S., 2017. Manure nitrogen production and application in cropland and rangeland during 1860–2014: A 5-minute gridded global data set for Earth system modeling. *Earth Syst. Sci. Data Discuss.* 1–35. <https://doi.org/10.5194/essd-2017-11>

REVIEWERS' COMMENTS

Reviewer #1 (Remarks to the Author):

I am satisfied with the answers the authors provided to my comments in the rebuttal letter (and also to the comments of the other two reviewers).

I am also satisfied with the revisions and additional analyses that the authors made, which made the manuscript stronger.

I have no further comments.

Reviewer #2 (Remarks to the Author):

Thank you very much for the thorough response to the reviewers. Almost all comments were taken into account, and I very much appreciate the resulting new paper.

I do have one very small comments left, which is not critical for publication:

My first comment was

R2C2: p. 3, line 67: The authors mention that empirical 'nutrient based' VCmax models use PFT specific linear equations to calculate VCmax from LNC, but how do they get LNC? From data? From RS, or an internal model result? This matters quite a bit, and data is not very much available.

The response by the authors is: We apologize for the confusion. We used a published global leaf nitrogen content (LNC; g/m²) map – EB17 (Butler et al., 2017) in our analysis, which is upscaled from in-situ observations using plant functional maps, climate and soil variables.

But I thought (and still think) that the authors in this sentence are referring not to their analysis, but to how in 'nutrient based' VCmax models VCmax is calculated from LNC using linear equations specific for PFTs? The authors already explain well what they use for LNC map, but my question was related to the 'nutrient based' VCmax models.

Reviewer #3 (Remarks to the Author):

I want to congratulate the authors for addressing my main concerns and for developing such an interesting analysis of global variation in fLNR and Vcmax.

Said that, I still think that the authors have to support better the final decision on using the EB17 maps among others. As an example, Table S3 shows a marginal improvement in R² for EB17 in comparison with AMM18 (the reason to use the EB17 according to the authors), but the AMM18 map has a significant improvement in terms of RMSE. Moreover, the AMM18 map seems to be more consistent with both EB17 and CB20 (r values in subfigures d,e,f).

Just a couple of things that could be improved and are related with my concerns of the first review:

I think there could be a wrong statement in the manuscript that needs to be further checked: “and the weighting strategy of EB17 (i.e., unweighted by species abundance within a pixel) is close to the weighting strategy we adopted for V_{cmax} estimation”. I think the weighting strategies of EB17 and AMM18 are basically same, they are corrected from PFT biases in species sampling in both cases.

About the statement: “The accuracy of the conventional cross-validation is comparable to previous trait upscaling studies (Boonman et al., 2020; Butler et al., 2017; Moreno-Martínez et al., 2018; Van Bodegom et al., 2014)”. But what about the S.C.V? According to figure S11, the S.C.V. RMSE almost doubles the C.V. one. I think the results deserve a comment about some possible reasons for that and perhaps a recommendation to the final user about the significant increase in uncertainty due to this.

Global variation in the fraction of leaf nitrogen allocated to photosynthesis

NCOMMS-21-01819A

Response to reviewers

Authors: We thank the reviewer for their constructive comments, which have helped us greatly improve the manuscript. We appreciate the editor's decision to in-principle publish our study. The editorial requests have been answered in the cover letter. Here are our point-by-point response to the remaining reviewers' comments.

REVIEWERS' COMMENTS

Reviewer #1 (Remarks to the Author):

I am satisfied with the answers the authors provided to my comments in the rebuttal letter (and also to the comments of the other two reviewers).

I am also satisfied with the revisions and additional analyses that the authors made, which made the manuscript stronger.

I have no further comments.

Authors: We thank the reviewer for supporting our study and appreciate their constructive comments in the process.

Reviewer #2 (Remarks to the Author):

Thank you very much for the thorough response to the reviewers. Almost all comments were taken into account, and I very much appreciate the resulting new paper.

Authors: We are grateful to the reviewer for their detailed comments and glad to know we have addressed most of the comments.

I do have one very small comments left, which is not critical for publication:

My first comment was

R2C2: p. 3, line 67: The authors mention that empirical 'nutrient based' VCmax models use PFT specific linear equations to calculate VCmax from LNC, but how do they get LNC? From data?

From RS, or an internal model result? This matters quite a bit, and data is not very much available.

The response by the authors is: We apologize for the confusion. We used a published global leaf nitrogen content (LNC; g/m²) map – EB17 (Butler et al., 2017) in our analysis, which is upscaled from in-situ observations using plant functional maps, climate and soil variables.

But I thought (and still think) that the authors in this sentence are referring not to their analysis, but to how in ‘nutrient based’ Vcmax models Vcmax is calculated from LNC using linear equations specific for PFTs? The authors already explain well what they use for LNC map, but my question was related to the ‘nutrient based’ Vcmax models.

Authors: Thank you for the comment. Sorry we misunderstood the question. The statement mentioned by the reviewer introduced a type of Vcmax model that is based on leaf nutrient. We fully agree that the leaf nutrient data from multiple sources have been used (i.e., observed *in-situ* data (Kattge et al., 2009), internal model results(Walker et al., 2017)) or could be used (i.e., data-driven LNC map) to drive the nutrient-based Vcmax models.

As the main purpose of this statement in the Introduction is to showcase the underlying assumption of fLNR for nutrient-based Vcmax models (e.g., almost constant fLNR), regardless of their nutrient data input. Therefore, we did not expand on the technical details of which data sources were used by previous studies. We have updated a statement in the Methods (L538) to highlight that we used the data-driven LNC map EB17 to drive the multiple nutrient-based Vcmax models (EM1, EM2, EM3 and EM4) in our study.

Reviewer #3 (Remarks to the Author):

I want to congratulate the authors for addressing my main concerns and for developing such an interesting analysis of global variation in fLNR and Vcmax.

Authors: We appreciate the constructive comments from the reviewer and glad to know the main concerns have been addressed.

Said that, I still think that the authors have to support better the final decision on using the EB17 maps among others. As an example, Table S3 shows a marginal improvement in R² for EB17 in comparison with AMM18 (the reason to use the EB17 according to the authors), but the AMM18 map has a significant improvement in terms of RMSE. Moreover, the AMM18 map seems to be more consistent with both EB17 and CB20 (r values in subfigures d,e,f).

Authors: Thank you for the comment. We agree with the reviewer that AMM18 has lower RMSE than EB17 and EB17 has slightly better r² than AMM18. However, regarding their consistence with other datasets, our result (Supplementary Figure 9d,e,f) in fact shows EB17 is

more consistent with other datasets (EB17 vs AMM18 $r = 0.51$, EB17 vs CB20 $r = 0.28$) than AMM18 (AMM18 vs EB17 $r = 0.51$, AMM18 vs CB20 $r = 0.17$). In light of this, we think it is hard to argue AMM18 is better than EB17, or vice versa.

After all, we have showed that the choice between AMM18 and EB17 is an issue of limited importance to our conclusion, as the dominant controls for fLNR are the same when using either one for analysis (Supplementary Figure 10). We have added the following statement in the Discussion (L331-337).

“In our study, we used EB17 as the principal LNC map since it demonstrated the highest R^2 in validation (Supplementary Table 3) and it was more consistent with the other two LNC maps than AMM18 (Supplementary Figure 9). We acknowledge that the AMM18 LNC map has a smaller RMSE in validation compared to EB17 though it has slightly lower R^2 (Supplementary Table 3). In this study, we do not identify which LNC map is more accurate, but show that the choice between EB17 and AMM18 has a limited influence on our conclusion regarding the dominant controls for fLNR (Supplementary Figure 10g, h).”

We have also generated multiple versions of fLNR based on different LNC datasets (i.e., EB17, AMM18, CB20) (<https://zenodo.org/record/5090497>), so readers can use either one at their discretion.

Just a couple of things that could be improved and are related with my concerns of the first review:

I think there could be a wrong statement in the manuscript that needs to be further checked: “and the weighting strategy of EB17 (i.e., unweighted by species abundance within a pixel) is close to the weighting strategy we adopted for Vcmax estimation”. I think the weighting strategies of EB17 and AMM18 are basically same, they are corrected from PFT biases in species sampling in both cases.

Authors: Thank you for pointing that out. We have removed the statement following your suggestion.

After a double check, we realized that EB17, AMM18 and our study used different weight strategies. EB17 considered all species equally in a pixel (though they tested weights using TRY sampling abundance), AMM18 used high resolution remote sensing PFT and species characteristics to get PFT weights for pixels, while our study did not consider species weights.

About the statement: “The accuracy of the conventional cross-validation is comparable to previous trait upscaling studies (Boonman et al., 2020; Butler et al., 2017; Moreno-Martínez et al., 2018; Van Bodegom et al., 2014)”. But what about the S.C.V? According to figure S11, the S.C.V. RMSE almost doubles the C.V. one. I think the results deserve a comment about some

possible reasons for that and perhaps a recommendation to the final user about the significant increase in uncertainty due to this.

Authors: Thanks for the comment. We added a new statement to acknowledge the result on spatial cross validation(L492-496).

“The spatial cross-validation showed more variable R^2 and larger RMSE than the conventional cross-validation as the former has less samples for validation after removing the spatially autocorrelated samples, and those spatially autocorrelated samples generally have small RMSEs between the observations and the estimates”.

Kattge, J., Knorr, W., Raddatz, T., Wirth, C., 2009. Quantifying photosynthetic capacity and its relationship to leaf nitrogen content for global-scale terrestrial biosphere models. *Glob. Chang. Biol.* 15, 976–991. <https://doi.org/10.1111/j.1365-2486.2008.01744.x>

Walker, A.P., Quaife, T., van Bodegom, P.M., De Kauwe, M.G., Keenan, T.F., Joiner, J., Lomas, M.R., MacBean, N., Xu, C., Yang, X., Woodward, F.I., 2017. The impact of alternative trait-scaling hypotheses for the maximum photosynthetic carboxylation rate (V_{cmax}) on global gross primary production. *New Phytol.* 215, 1370–1386. <https://doi.org/10.1111/nph.14623>